

# The Bulimulidae (Mollusca: Pulmonata) from the Región de Atacama, northern Chile

Juan Francisco Araya

Universidad de Atacama, Copiapó, Región de Atacama, Chile
Programa de Doctorado en Sistemática y Biodiversidad, Universidad de Concepción,
Concepción, Chile

## ABSTRACT

The bulimulid genus *Bostryx Troschel, 1847* is the most species-rich genus of land snails found in Chile, with the majority of its species found only in the northern part of the country, usually in arid coastal zones. This genus has been sparsely studied in Chile and there is little information on their distribution, diversity or ecology. Here, for the first time, a formal analysis of the diversity of bulimulids in the Región de Atacama, northern Chile, is reported. Of the seventeen species recorded for the area, most of them were efectively found in the field collections and one record was based on literature. Five taxa are described as new: *Bostryx ancavilorum* sp. nov., *Bostryx breurei* sp. nov., *Bostryx calderaensis* sp. nov., *Bostryx ireneae* sp. nov. and *Bostryx valdovinosi* sp. nov., and the known geographic distribution of seven species is extended. Results reveal that the Región de Atacama is the richest region in terrestrial snails in Chile, after the Juan Fernández Archipelago. All of the terrestrial molluscan species occurring in the area are endemic to Chile, most of them with restricted geographic distributions along the coastal zones, and none of them are currently protected by law. Further sampling in northern Chile will probably reveal more snail species to be discovered and described.

## INTRODUCTION

Land snail diversity is only surpassed by the Arthropoda diversity (*Cowie & Robinson, 2003*), they have a larger index of recent extinctions compared to any other taxa (*Régnier, Fontaine & Bouchet, 2009*) and thus the knowledge of their taxonomic classification and distribution is essential in biodiversity studies. In Chile, land snails are one of the least studied molluscan groups; apart from the early works describing species in the ninetenth century (*Broderip & Sowerby, 1832a*; *Broderip & Sowerby, 1832b*; *King & Broderip, 1832*; *Sowerby, 1833*; *D'Orbigny, 1834–1847*; *D'Orbigny, 1835*; *Anton, 1838*; *Pfeiffer, 1842*; *Pfeiffer, 1847*; *Souleyet, 1842*; *Reeve, 1848–1850*; *Hupé, 1854*, among others), and of a few works in the twentieth century (*Odhner, 1922*; *Odhner, 1963*; *Gigoux, 1932*; *Rehder, 1945*; *Biese, 1949*; *Biese, 1960*; *Herm, 1970*; *Tillier, 1981*), the last major revision listed 154 species of terrestrial molluscs occurring in the country (*Stuardo & Vega, 1985*). Further

Corresponding author
Juan Francisco Araya,
jfaraya@u.uchile.cl

works only include a bibliographical publication on Bulimulidae (*Stuardo & Valdovinos, 1985*), a detailed revision of the genus *Plectostylus* (*Valdovinos & Stuardo, 1988*), the revision of the families Veronicellidae, Pupillidae and Achatinellidae occurring in the country (*Stuardo & Vargas-Almonacid, 2000*), a study of the endemic slug *Phyllocaulis gayi* (*Fischer, 1871*) (*Simonetti, Grez & Bustamante, 2003*), of a charopid and an endemic acavid species (*Cádiz & Gallardo, 2008*; *Silva & Thomé, 2009*), and the description of a few new species, all of them charopid or endodontid micromollusks (*Valdovinos & Stuardo, 1989*; *Vargas-Almonacid, 2000*; *Vargas-Almonacid & Stuardo, 2007*; *Miquel & Barker, 2009*; *Miquel & Cádiz-Lorca, 2008*; *Araya & Aliaga, 2015a*), and a sub fossil Achatinellid from Easter Island (*Kirch, Christensen & Steadman, 2009*). Land Mollusca from northern Chile in particular were studied in greatest depth by Pilsbry in his monumental treatise of the pulmonate Mollusca (*Pilsbry, 1895–1896*). Subsequent studies only include *Philippi (1860)* and *Gigoux (1932)* citing the species living in the Región de Atacama, *Rehder (1945)*, who reviewed the subgenus *Peronaeus*, describing two new species, and the works of *Breure (1978)*, *Valdovinos & Stuardo (1988)*, Miquel & *Araya (2013)*, *Araya & Catalán (2014)* and *Araya (2015)*, the latter reviewing the non-indigenous species found in the country.

The family Bulimulidae Tryon, 1867 is represented in Chile solely by the genus *Bostryx* Troschel, 1843, an endemic South American genus distributed from Suriname to Chile (*Breure, 1978*). In Chile, this genus is found mostly on coastal localities in northern-central Chile, from Arica, Región de Arica y Parinacota (18°29′S, 70°20′W), to Coquimbo, Región de Coquimbo (29°57′S, 71°20′W). This work presents a review, with distributions and illustrations, of all the *Bostryx* species found in the Región de Atacama, northern Chile, including a key to all the taxa under consideration. The aim of this preliminary paper is to contribute to the knowledge of the land snail fauna in Chile, particularly to assess conservation status of all these species, which are particularly endangered in the coastal areas of the country.

## MATERIALS AND METHODS

The Región de Atacama is situated in northern Chile (25°17′S to 29°30′S), it covers approximately 75,176 km$^2$ and has one of the lowest human population densities in the country, being also one of the most desertic areas of Chile. This Región occupies the southern part of the Atacama Desert and has a cold desert climate (BWk) (*Peel, Finlayson & McMahon, 2007*), and a high relative humidity (Average 74%) in its coastal areas, product of night fogs and high coastal nubosity; with very scarce precipitation, most of which is associated with the El Niño Southern Oscillation (ENSO) events (*Juliá, Montecinos & Maldonado, 2008*; *Squeo et al., 2006*). Detailed descriptions of the surveyed area, particularly of the flora and higher fauna are provided in *Squeo, Arancio & Gutiérrez (2008)*.

Most of the collections were made in the coastal desert areas around the port of Caldera (27°04′S; 70°50′W) during the summers of 2009 to 2012 and in August–December 2012. Surveys were undertaken in a similar manner as those of *Cowie & Robinson (2003)* and also by collecting litter for further sorting in the laboratory. A synopsis of the localities is

**Table 1 Table of collecting localities and its diversity of species.** Collecting localities of terrestrial mollusca in the Región de Atacama, arranged according to latitude from north to south. Coordinates denote one representative point of many stations for each locality. Occurring species involve also species cited by Miquel & *Araya (2013)* and by *Araya & Catalán (2014)*.

| Locality | Coordinates/altitude | Ecology | Ocurring species |
|---|---|---|---|
| Obispito | 26°44′S; 70°42′W. | Coastal plains, moderate vegetation. | *B. mejillonensis.* |
| Aguas Verdes | 26°52′S; 70°48′W, 60 m. | Low coastal hills with rocky outcrops, scarce vegetation. | *B. albicans, B. ancavilorum* sp. nov. |
| Zoológico de Piedra | 26°56′S; 70°47′W, 94 m. | Rocky outcrop with sparse vegetation. | *Plectostylus broderipii.* |
| Quebrada del León | 26°57′S; 70°44′W, 378 m (Hill). 26°58′S; 70°45′W, 155 m. | Sandy plains and rocky hills with moderate vegetation of cacti and desert herbs. | *B. breurei* sp. nov., *Stephacharopa calderaensis.* |
| Caldera Bay | 27°04′S; 70°49′W, 54 m. | Sandy plains with very scarce vegetation. | *B. calderaensis* sp. nov., *Plectostylus broderipii.* |
| Plains NE Caldera. | 27°04′22″S; 70°49′3″W, 135 m. | Coastal plain, almost no vegetation and Rocky hills with scarce vegetation. | *B. albicans.* |
| El Morro Hill | 27°08′43″S; 70°55′42″W, 194 m. | Steep rocky terrain near the sea, moderate vegetation of herbs and cacti, plentiful lichen communities. | *B. valdovinosi* sp. nov., *B. ischnus, B. pumilio, Plectostylus broderippii, Plectostylus coturnix, Stephacharopa calderaensis.* |
| Copiapó | 27°21′29″S; 70°20′24″W, 470 m. | Small mountains, very scarce vegetation. | *B. inaquosum* |
| La Virgen beach Area | 27°21′29″S; 70°20′24″ W, 55 m. | Hills near the sea, moderate vegetation of herbs, scarce cacti. | *B. huascensis.* |
| Barranquilla beach Area | 27°21′29″S; 70°20′24″W, 123 m. | Sandy plains and rocky outcrops with scarce vegetation. | *B. erythrostomus.* |
| Huasco | 28°30′46″S; 71°12′25″W, 171 m. | Sandy plains with moderate vegetation of herbs. | *B. pustulosus, B. rhodacme, B. ischnus.* |
| Chañaral de Aceituno | 29°01′35″S; 71°26′ 20″W, 174 m | Sandy hills with scarce vegetation. | *B. ireneae* sp. nov., *B. pustulosus.* |

given in Table 1. The studied samples were variably preserved, from live specimens with periostracum and epiphragm to eroded shells. The terminology of shell morphology is based upon *Breure (1979)* and the measurements follow *Breure & Ablett (2012)* for the whorl counts. The literature and specially the original description of each species were carefully reviewed, and the references included in the synonymies are mostly the ones that contained detailed descriptions or figures. Field study permits were not required for this study and none of the species studied herein are currently under legal protection. The dimensions of the shells, measured with Vernier calipers (±0.1 mm) are depicted in Fig. 1. Abbreviations used are: MZUC, Museo de Zoología de la Universidad de Concepción, Concepción, Chile; RMNH.MOL, Naturalis Biodiversity Centre, The Netherlands, Mollusca collection; MPCCL, Museo Paleontológico de Caldera, Caldera, Chile; RCGCL, Private collection of Ricardo Catalán Garrido, Copiapó, Chile; USNM, Smithsonian Institute National Museum of Natural History Washington, USA and JFACL, reference collection of the author (Caldera, Chile). Other abbreviations used are: *D*, diameter of the shell: maximum dimension perpendicular to *H*, including lip; *H*, shell height: maximum dimension parallel to axis of coiling, including lip; *HA*, height of aperture, including

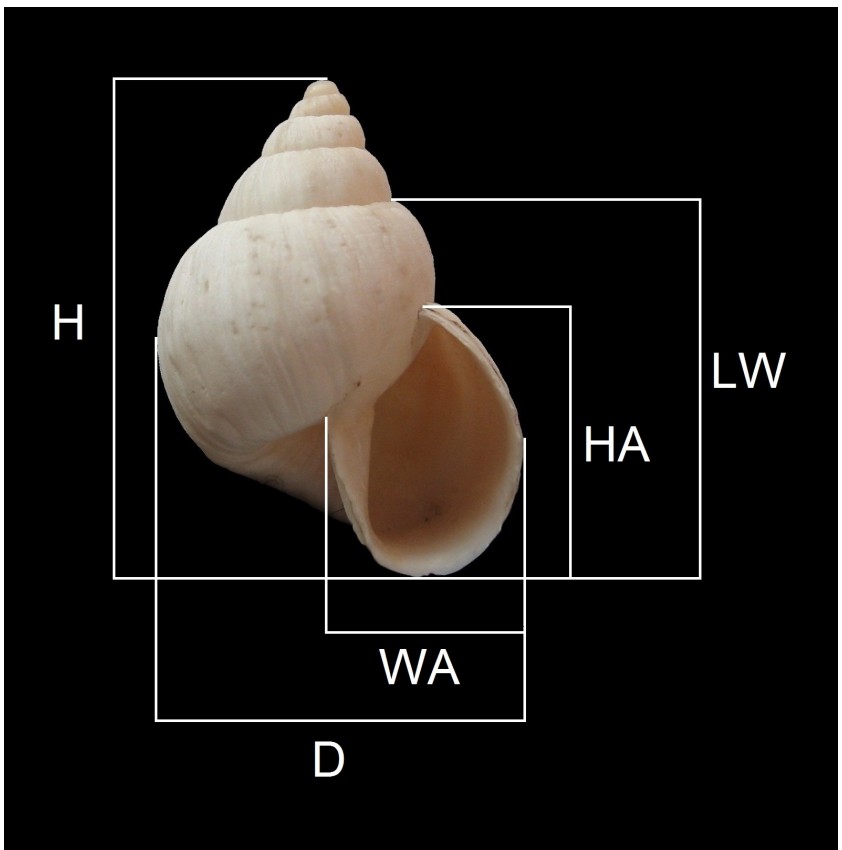

**Figure 1 Measurements performed on shells.** Abbreviations are: *D*, width of shell (maximum dimension perpendicular to *H*, including lip); *H*, shell height (maximum dimension parallel to axis of coiling, including lip); *HA*, height of aperture (including lip); *LW*, height of last whorl (including lip); *WA*, width of aperture (including lip).

lip; *HT*, holotype; *LW*, height of last whorl, including lip; *NW*, number of whorls; *PT*, paratype *WA*, width of aperture, including lip. All the measurements are given in mm, otherwise indicated.

The electronic version of this article in Portable Document Format (PDF) will represent a published work according to the International Commission on Zoological Nomenclature (ICZN), and hence the new names contained in the electronic version are effectively published under that Code from the electronic edition alone. This published work and the nomenclatural acts it contains have been registered in ZooBank, the online registration system for the ICZN. The ZooBank LSIDs (Life Science Identifiers) can be resolved and the associated information viewed through any standard web browser by appending the LSID to the prefix http://zoobank.org/. The LSID for this publication is: urn:lsid:zoobank. org:pub:3F2E9582-A5D0-431D-BC3E-3917DC812323. The online version of this work is archived and available from the following digital repositories: PeerJ, PubMed Central and CLOCKSS.

# RESULTS

## Systematic account

**Superfamily Orthalicoidea Martens in** *Albers, 1860*
**Family Bulimulidae Tryon, 1867**
**Subfamily Bostrycinae** *Breure, 2012*
**Genus** ***Bostryx*** *Troschel, 1847*

**Type species** *Bulimus (Bostryx) solutus* *Troschel, 1847*, by monotypy. Recent, Lima, Perú.

### *Bostryx albicans* (**Broderip, 1832**)

(Figs. 2A–2K)
*Bulinus albicans* Broderip, 1832: 105; Hupé *in* Gay, 1854: 109, pl. 3, Fig. 6. *Bulimulus* (*Lissoacme*) *albicans* Pilsbry, 1896: 175, pl. 48, Figs. 4–5. *Bostryx* (*Lissoacme*) *albicans* *Stuardo & Valdovinos, 1985*: 56; *Stuardo & Vega, 1985*: 135. *Bostryx* (*Peronaeus*) *albicans* *Valdovinos, 1999*: 150.

**Material examined:** Cerro Caracoles (26°58′23″S; 70°43′52″W), about 15 km NE of the port of Caldera, Región de Atacama, Chile; leg. by Marta Araya E, dec. 2013 (JFACL TG 015, 8 sppm; MZUC 39623, 1 spm; MZUC 39622, 1 spm). Measurements of illustrated sppm (in mm): Figs. 2A–2F MZUC 39623 ($H = 21.3$, $D = 14.0$, $LW = 17.0$, $HA = 10.7$, $WA = 8.3$, $NW = 5.5$); Figs. 2G–2K MZUC 39622 ($H = 18.7$, $D = 13.3$, $LW = 16.2$, $HA = 10.8$, $WA = 9.2$, $NW = 4.5$).

**Description (After Pilsbry, 1896):** Shell umbilicated, obese-ovate, rather thin; with five and a half convex white whorls with indistinct grayish streaks or closely speckled and streaked throughout with brown, the markings translucent by transmitted light. Apex roseate or corneous; protoconch smooth (at high magnification with very fine lines near the sutures), translucent, yellowish white or roseate in juvenile specimens. Spire conic, apex rather obtuse, smooth. Surface with rather coarse, irregular growth-wrinkles, more or less plicate below sutures, and decussated above the middle by spiral incised lines, rather few and sometimes subobsolete. Aperture over half the height, ovate, white to brownish inside; outer lip thin, sharp, unexpanded. Columella dilated above, nearly straight, brownish or rose; parietal wall with almost imperceptible glaze.

**Distribution:** *Stuardo & Vega (1985)* cited this species from Copiapó, Región de Atacama to Coquimbo, Región de Coquimbo. This is the northernmost record for the species.

**Remarks:** The conspicuous roseate-red apex is more notable in juvenile snails (Fig. 2G), as the pink hue fades quickly to pale yellowish or white (Fig. 2E). This is one of the most abundant land snail species in the Región de Atacama, but it is restricted to sandy locations in hills of the Chilean Coastal Range.

### *Bostryx ancavilorum* **sp. nov.**

Urn:lsid:zoobank.org:act:79735125-EE09-4FEF-B1D1-E86810CF7372

(Figs. 2L–2V)

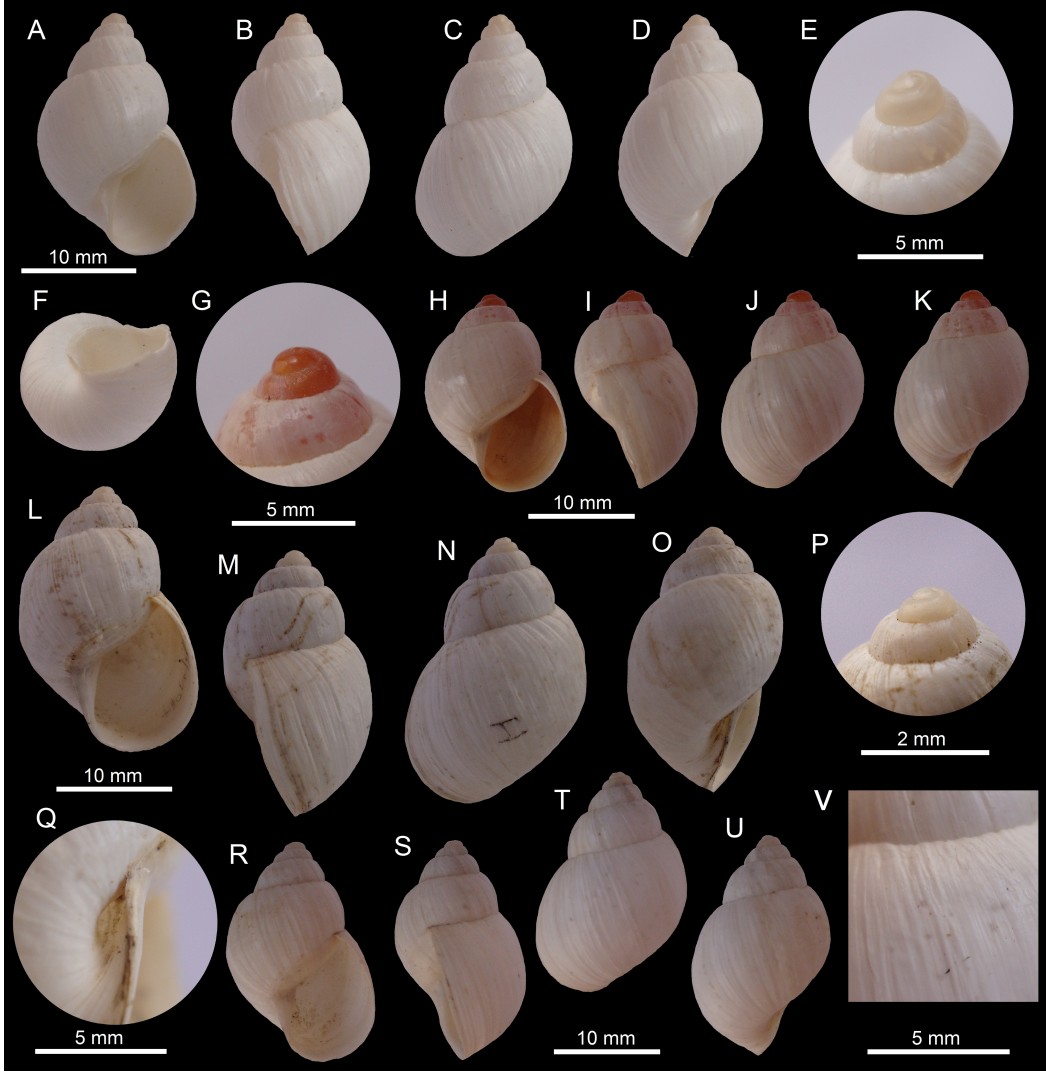

**Figure 2 Images of shells of *Bostryx* species.** *Bostryx albicans* (Broderip, 1832) MZUC 39623 (*H*: 21.3 mm), (A) ventral view, (B) side view (lip), (C) dorsal view, (D) side view (umbilicus), (E) detail of protoconch, (F) umbilical view; *Bostryx albicans* (Broderip, 1832) juvenile specimen MZUC 39622 (H: 18.7 mm), (G) detail of protoconch, (H) ventral view, (I): side view (lip), (J) dorsal view, (K) side view (umbilicus); *Bostryx ancavilorum* sp. nov. HT MPCCL 13102015K (*H*: 23.2 mm), (L) ventral view, (M) side view (lip), (N) dorsal view, (O) side view (umbilicus), (P) detail of protoconch, (Q) detail of umbilicus; *Bostryx ancavilorum* sp. nov. PT 1 MPCCL 13102015E (*H*: 20.9 mm), (R) ventral view, (S) side view (lip), (T) dorsal view, (U) side view (umbilicus), (V) detail of sculpture.

**Type material:** Holotype MPCCL 13102015K ($H = 23.2$, $D = 15.9$, $LW = 19.4$, $HA = 14.0$, $WA = 10.2$, $NW = 5.0$), paratype 1 MPCCL 13102015E ($H = 20.9$, $D = 14.2$, $LW = 17.2$, $HA = 12.0$, $WA = 10.0$, $NW = 5.0$), paratype 2 MZUC 39621 ($H = 22.7$, $D = 15.4$, $LW = 18.6$, $HA = 13.0$, $WA = 10.4$, $NW = 5.5$). All the material collected at the type locality.

**Type locality:** Aguas Verdes (26°52′S; 70°48′W, 60 m), Comuna de Caldera, Región de Atacama, Chile; collected by JF Araya, February 12, 2012. All the specimens were collected in sand among rocks.

**Diagnosis:** A *Bostryx* species characterized by a thin, obese elongate white shell, with slightly shouldered whorls, aperture height 0.60 of shell heigth and a corneous proto-conch.

**Description:** Shell up to 23.5 mm, around 1.51 times as long as wide, stout, obese-elongate, with convex sides, somewhat shouldered under sutural area, umbilicus narrow. Shell white, surface shiny with marked growth striae. Protoconch smooth, slightly wrinkled near sutures. Whorls 5.25–5.50, convex, last whorl very slightly appresed at periphery, compressed below, suture well-impressed and slightly plicate. Aperture ovate, around 1.34 times as long as wide, aperture height around 0.59 the total height. Peristome simple, sharp, narrowly expanded in its anterior side. Columellar margin slightly concave, folded over the umbilicus, well dilated above. Parietal callus thin, shiny and transparent, with slightly nacreous glaze. Soft parts unknown.

**Distribution:** Only known from type locality.

**Etymology:** Named in honour of Juan Antonio Aliaga Ancavil (Universidad de Chile, Santiago, Chile) and family, for his friendship and support.

**Remarks:** This species was found in sand among rocks; its shell is somewhat similar to *B. albicans* (Broderip, 1832), from which it differs in having a larger and thinner shell, with a more developed parietal callus, a sharper lip and a comparatively larger and rounder aperture (specially different at its anterior side). It differs from *B. pruinosus* (Sowerby I, 1833) in having a larger, thinner and wider shell with a smaller umbilicus, more convex spire and a smaller last whorl. The species *B. albus* (Sowerby I, 1833) differs in having a stouter, smaller shell, with a more rounded aperture and a yellowish-white or light brown parietal callus.

### *Bostryx breurei* sp. nov.

Urn:lsid:zoobank.org:act:312AEEF0-C859-4C26-BEBD-8C03CA664A10

(Figs. 3A–3J)

**Type material:** Holotype MPCCL 13102015A ($H = 17.3$, $D = 8.9$, $LW = 12.7$, $HA = 8.8$, $WA = 5.1$, $NW = 6.0$), paratype MZUC 39620 ($H = 16.6$, $D = 8.6$, $LW = 12.7$, $HA = 8.6$, $WA = 5.0$, $NW = 6.0$). All the material collected at the type locality.

**Type locality:** Quebrada del León (26°57′29″S; 70°44′22″W, 423 to around 500 m), Comuna de Caldera, Región de Atacama, Chile; collected by JF Araya, January 15, 2012 and collected by R Catalán, December 2010. Both specimens, and fragments of shells, were collected under the roots of dry cacti, among rocks.

**Diagnosis:** A *Bostryx* species characterized by an elongate-ovate fragile thin shell, with an acute conic spire, a prominent last whorl which encompasses 0.6 of the shell heigth and a conspicuous shell sculpture of crossed axial and spiral lines.

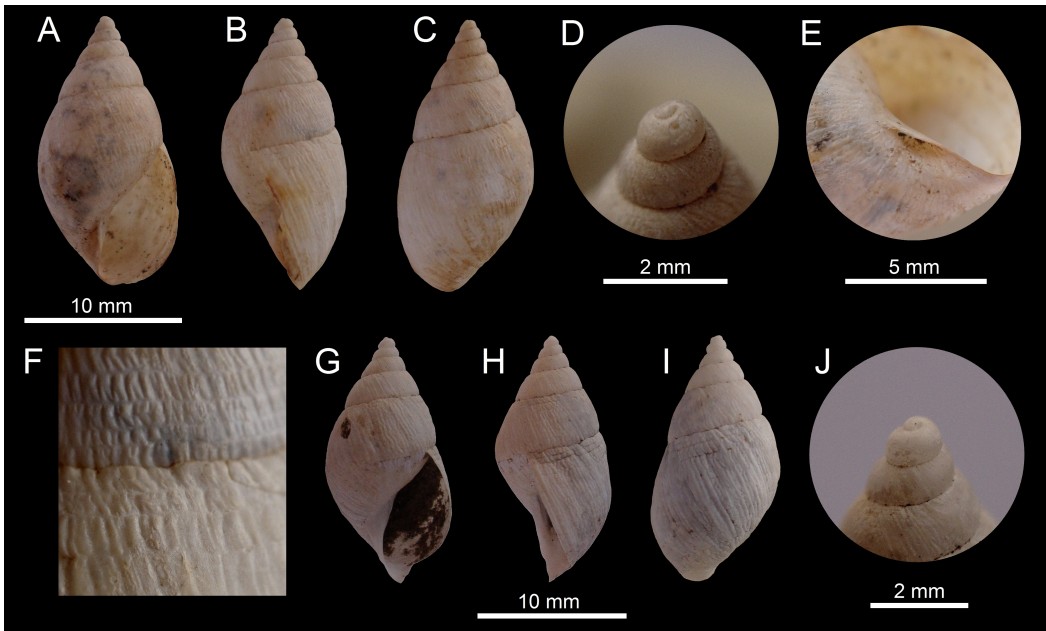

**Figure 3** **Images of shells of *Bostryx* species.** *Bostryx breurei* sp. nov. HT MPCCL 13102015A (*H*: 17.2 mm), (A) ventral view, (B) side view (lip), (C) dorsal view, (D) detail of protoconch, (E) detail of umbilicus, (F) detail of sculpture and suture; *Bostryx breurei* sp. nov. PT MZUC 39620 (*H*: 16.6 mm), (G) ventral view, (H) side view (lip), (I) dorsal view, (J) detail of protoconch.

**Description:** Shell up to 16.8 mm, 1.95 times as long as wide, elongate-ovate, rimate-umbilicate, with slightly convex whorls, fragile. Shell thin, translucent, slightly pinkish, decorated with minute streaks of opaque white. Surface lustrous, with marked axial growth striae, fine spiral striae more marked towards the subsutural area, where it crosses the axial growth lines. Protoconch small, smooth; on magnification it shows a fine micro-sculpture of axial costulae. Spire acute, slightly concave, with 6.25 slightly convex whorls, a little angulated near the well-impressed sutures. Last whorl very slightly carinated at the perifery, compressed below. Aperture elongate-subovate, 1.77 times as long as wide, 0.55 times the total height. Peristome simple, sharp, very fragile. Columellar margin concave and folded over the shallow umbilicus, well dilated above. Soft parts unknown.

**Distribution:** Only known from type locality.

**Etymology:** In honour of Abraham Breure (Naturalis Biodiversity Centre, Leiden, The Netherlands), for his contributions to the taxonomy of land Mollusca, especially that of Chile and Perú, and in particular for his extensive work on Bulimulidae.

**Remarks:** This new species has a distinctive light and fragile shell, different from all the other *Bostryx* species found in the area. *Bostryx paposensis* (*Pfeiffer, 1856*), described from Paposo (25°05′S; 70°25′W), in the Región de Antofagasta, has a shell of similar outline, however its protoconch and first whorls are smaller and darker, the whorls are almost flat and the spire is more convex. This is a rare species, from which only two, somewhat worn complete specimens (paratype with a broken external lip; Fig. 3G) were recovered from

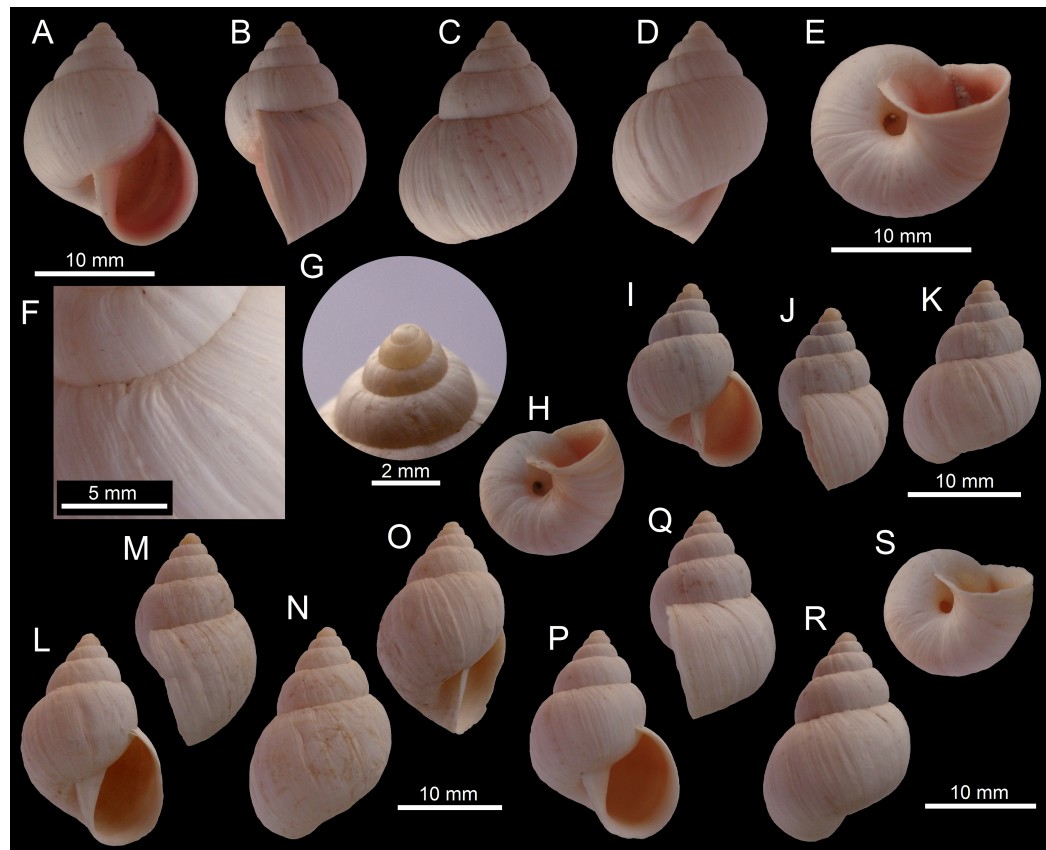

**Figure 4 Images of shells of *Bostryx* species.** *Bostryx calderaensis* sp. nov. HT MPCCL 13102015J (*H*: 18.9 mm), (A) ventral view, (B) side view (lip), (C) dorsal view, (D) side view (umbilicus), (E) umbilical view, (F) detail of sculpture and suture, (G) detail of protoconch; *Bostryx calderaensis* sp. nov. PT 1 MZUC 39619 (*H*: 16.3 mm), (H) umbilical view, (I) ventral view, (J) side view (lip), (K) dorsal view, PT 2 MPCCL 13102015L (*H*: 20.5 mm), (L) ventral view, (M) side view (lip), (N) dorsal view, (O) side view (umbilicus), PT 3 MZUC 39618 (*H*: 18.9 mm), (P) ventral view, (Q) side view (lip), (R) dorsal view, (S) umbilical view.

a single locality (Quebrada del León), among large rocks under cacti debris, in the top of mountains facing the sea in the Chilean Coastal Range.

***Bostryx calderaensis* sp. nov.**

Urn:lsid:zoobank.org:act:F54AAAB0-830B-41FA-B7D1-BC692083A590

(Figs. 4A–4S)

**Type material:** Holotype MPCCL 13102015J (*H* = 18.7, *D* = 14.5, *LW* = 14.6, *HA* = 11.2, *WA* = 8.8, *NW* = 5.5), paratype 1 MZUC 39619 (*H* = 16.3, *D* = 12.0, *LW* = 11.9, *HA* = 8.3, *WA* = 6.3, *NW* = 5.0), paratype 2 MPCCL 13102015L (*H* = 20.4, *D* = 13.7, *LW* = 15.7, *HA* = 11.1, *WA* = 8.5, *NW* = 5.5), paratype 3 MZUC 39618 (*H* = 18.7, *D* = 13.4, *LW* = 13.5, *HA* = 10.0, *WA* = 8.8, *NW* = 6.0). All material from type locality.

**Type locality:** Caldera (27°04′25″S; 70°49′09″W), Comuna de Caldera, Región de Atacama, Chile; collected by JF Araya, January 15, 2012. All the specimens were collected in sand, among large rocks.

**Diagnosis:** A *Bostryx* species characterized by a somewhat obese-ovate white shell, with a prominent last whorl, a pink to red colored inside of an oval-elongate aperture, with a strong and angulated columella and a deep umbilicus.

**Description:** Shell small (*H* up to 25 mm), obese-ovate, solid, opaque white, with 5.5–6.0 convex whorls; earlier whorls fleshy or corneous, smooth, separated by deep sutures. Protoconch large, smooth; in magnification with very fine wrinkles near the sutures. Surface of shell dull or slightly shining, with irregular wrinkles of growth. Spire slightly short, conic, last whorl slightly angulated at the periphery of shell. Aperture oval-elongate, slightly oblique, receding, over half the shell's length, pale pink to red within; lip moderately thick but sharp, slightly expanded, the edge whitish, simple. Columella broadly dilated, folded over the umbilicus; its inner edge straight or slightly concave, sharp; parietal callus a mere glaze. Umbilicus large, broad and deep. Soft parts unknown.

**Distribution:** Only known from type locality.

**Etymology:** Named after the type locality; Caldera, Región de Atacama, Chile.

**Remarks:** This new species has a distinctive deep umbilicus, which almost reach the protoconch (Fig. 4E), different from almost all of the other *Bostryx* species found in the area. This species is also slightly variable in morphology, with some specimens having shorter (Fig. 4A) or larger shells (Fig. 4L); the characteristic wide columella is, however, a constant feature in most of the specimens examined. *Bostryx erythrostomus* (Sowerby I, 1833) is somewhat similar to this new species, differing in the much larger shells, with a darker coloration of the aperture, the generally more convex whorls, a less pronounced umbilicus and in having a larger and shinier shell. *Bostryx huascensis* (Reeve, 1848) differs from the new species in its smaller shell with a larger spire and a comparatively smaller and pale brown to brownish colored aperture. This species is restricted to sandy areas around and in the city of Caldera and is exposed to urban development and human intervention of its habitat.

### *Bostryx erythrostomus* (Sowerby I, 1833)

(Figs. 5A–5P)

*Bulinus erythrostoma* Sowerby I, 1833: 37. *Bulimus erythrostoma* Reeve, 1849: Pl. XIII, Fig. 75; Hupé *in* Gay, 1854: 109, pl. 3, Fig. 3; *Hidalgo, 1870*: 85. *Bulimulus* (*Lissoacme*) *erythrostoma* Pilsbry, 1895: 173, pl. 48, Figs. 1–2, 97. *Bostryx* (*Lissoacme*) *erythrostomus*: *Stuardo & Valdovinos, 1985*: 56; *Stuardo & Vega, 1985*: 135. *Bostryx* (*Peronaeus*) *erythrostomus Valdovinos, 1999*: 150. *Bostryx erythrostoma Neubert & Janssen, 2004*: 209, pl. 6, Fig. 66; *Köhler, 2007*: 132, Fig. 26.

**Material examined:** Road to Barranquilla beach (27°21′29″S; 70°20′24″W), Comuna de Caldera, Región de Atacama, Chile; leg. by Ricardo Catalán G, December 2004 (RCGCL unnumbered, 34 sppm, JFACL TG 018) and coll. by Marta Esther Araya, July 2014 (MZUC 39617, 1 spm; JFACL TG018, 5 sppm). Measurements of illustrated sppm (in mm): Figs. 5A–5E JFACL TG018 ($H = 30.0$, $D = 26.4$, $LW = 25.3$, $HA = 19.5$, $WA = 16.4$, $NW = 5.5$); Figs. 5F–5K MZUC 39617 ($H = 25.7$, $D = 21.4$, $LW = 20.9$, $HA = 15.9$,

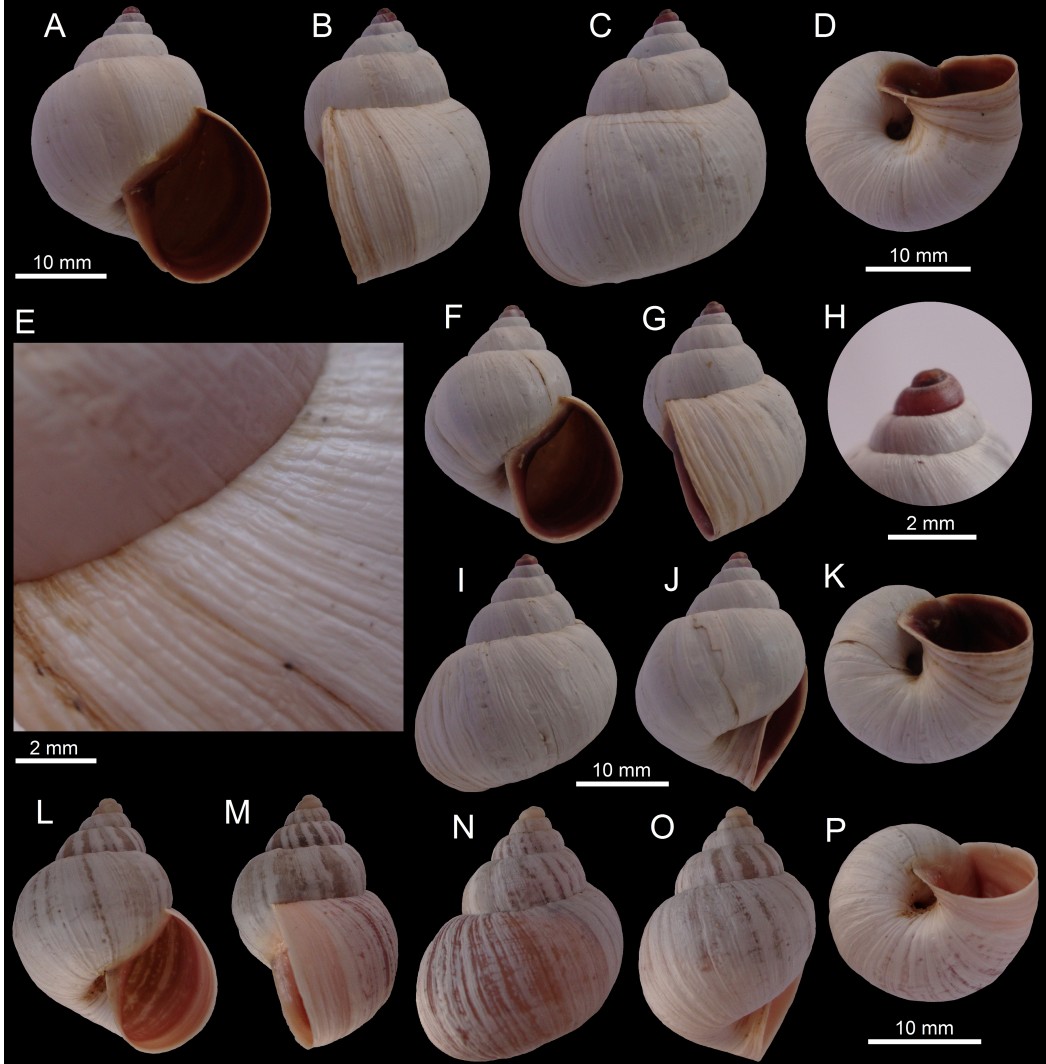

**Figure 5 Images of shells of *Bostryx* species.** *Bostryx erythrostomus* (Sowerby I, 1833) JFACL TG018 (*H*: 30.7 mm), (A) ventral view, (B) side view (lip), (C) dorsal view, (D) umbilical view, (E) detail of sculpture and suture; *Bostryx erythrostomus* (Sowerby I, 1833) MZUC 39617 (*H*: 25.9 mm), (F) ventral view, (G) side view (lip), (H) detail of protoconch, (I) dorsal view, (J) side view (umbilicus), (K) umbilical view; *Bostryx erythrostomus* (Sowerby I, 1833) juvenile specimen MPCCL 13102015I (*H*: 22.8 mm), (L) ventral view, (M) side view (lip), (N) dorsal view, (O) side view (umbilicus), (P) umbilical view.

$WA = 12.9$, $NW = 5.5$); Figs. 5l–5P MPCCL 13102015I ($H = 22.8$, $D = 18.2$, $LW = 17.8$, $HA = 12.8$, $WA = 9.9$, $NW = 5.0$).

**Description (After Pilsbry, 1896):** Shell obese-ovate, solid, white, with irregular indistinct fleshy or gray-blue streaks and small scattered dots, the latter translucent by transmitted light, these markings often inconspicuous; earliest whorls fleshy or corneous, smooth. Protoconch smooth, with very fine axial striae near the sutures visible only under magnification. Surface slightly shining, with irregular wrinkles of growth, and seen under the lens to be cut by superficial incised spiral lines into series of long granules, often absent on middle and base of last whorl. Spire short, conic, composed of 5.5–5.75

very convex whorls, separated by deep sutures. Last whorl rotund. Aperture slightly oblique, round-ovate, over half the shell's length, crimson, rose or red–brown within, becoming light brown in the throat; lip moderately thick but sharp, not expanded, the edge light in color. Columella broadly dilated, its inner edge straight or lightly concave; parietal callus thin, colored. Umbilicus large, colored inside.

**Distribution:** *Stuardo & Vega (1985)* recorded this species from Caldera and Huasco at the Región de Atacama, and from Coquimbo at the Región de Coquimbo.

**Remarks:** This species lives in sandy areas, among rocks; it has comparatively large obese shells ($H$ up to 30 mm in examined specimens; Figs. 5A–5E), of whitish, pinkish or even bluish white color, which are much more globose and with a much more conspicuous colored aperture than any other *Bostryx* species in the area under study. Juvenile specimens have corneous, variegated shells (Figs. 5L–5P). Of all the reviewed taxa found during this work, this is the only species represented by live specimens.

### *Bostryx huascensis* (Reeve, 1848)

(Figs. 6A–6F)

*Bulimus huascensis* Reeve, 1848: pl. 23, f. 147; *Hupé, 1854*: 111. *Bulimulus* (*Lissoacme*) *huascensis* Pilsbry, 1896: 174, pl. 48, Fig. 3. *Bostryx huascensis Breure, 1978*: 90. *Bostryx* (*Lissoacme*) *huascensis Stuardo & Valdovinos, 1985*: 56; *Stuardo & Vega, 1985*: 135. *Bostryx* (*Peronaeus*) *huascensis Valdovinos, 1999*: 150. *Bostryx huascensis* Breure & Ablett, 2014 : 90, Fig. 8B, L27ii.

**Material examined:** Near playa La Virgen (27°21′S; 70°57′W), Comuna de Caldera, Región de Atacama, Chile, coll. by JF Araya, February 2010 (JFACL TG014, 5 sppm; MPCCL 13102015M, 1 specimen). Measurements of illustrated spm (in mm): Figs. 6A–6F MPCCL 13102015M ($H = 19.6$, $D = 12.5$, $LW = 14.1$, $HA = 8.1$, $WA = 6.9$, $NW = 6.0$).

**Description (After Pilsbry, 1896):** Shell long-ovate, solid, opaque and white, with indistinct grayish streaks, or faintly pink with fleshy streaks. Surface with irregular, fine growth-wrinkles and subobsolete spiral incised lines above. Spire long, apex pink or corneous, obtuse, smooth. Whorls 6, convex, sutures deep. Aperture half as long as shell, ovate, light yellowish-brown inside; lip thin and acute. Columella broadly dilated, whitish, its inner edge straightened. Parietal wall with a light wash of white callus. Umbilicus narrow.

**Distribution:** *Stuardo & Vega (1985)* cited this species only from Huasco, Región de Atacama. This is the northernmost record for the species.

**Remarks:** This uncommon species looks like *B. albicans* in the general proportions of the shell. However, they differ in having a smaller (up to 19.5 mm), stouter shell with a more acute spire, and a larger umbilicus. *Bostryx calderaensis* sp. nov is also somewhat similar to this species, however it has larger shells, with a shorter spire and a larger aperture.

### *Bostryx inaquosum* *Breure, 1978*

(Figs. 6G–6I)

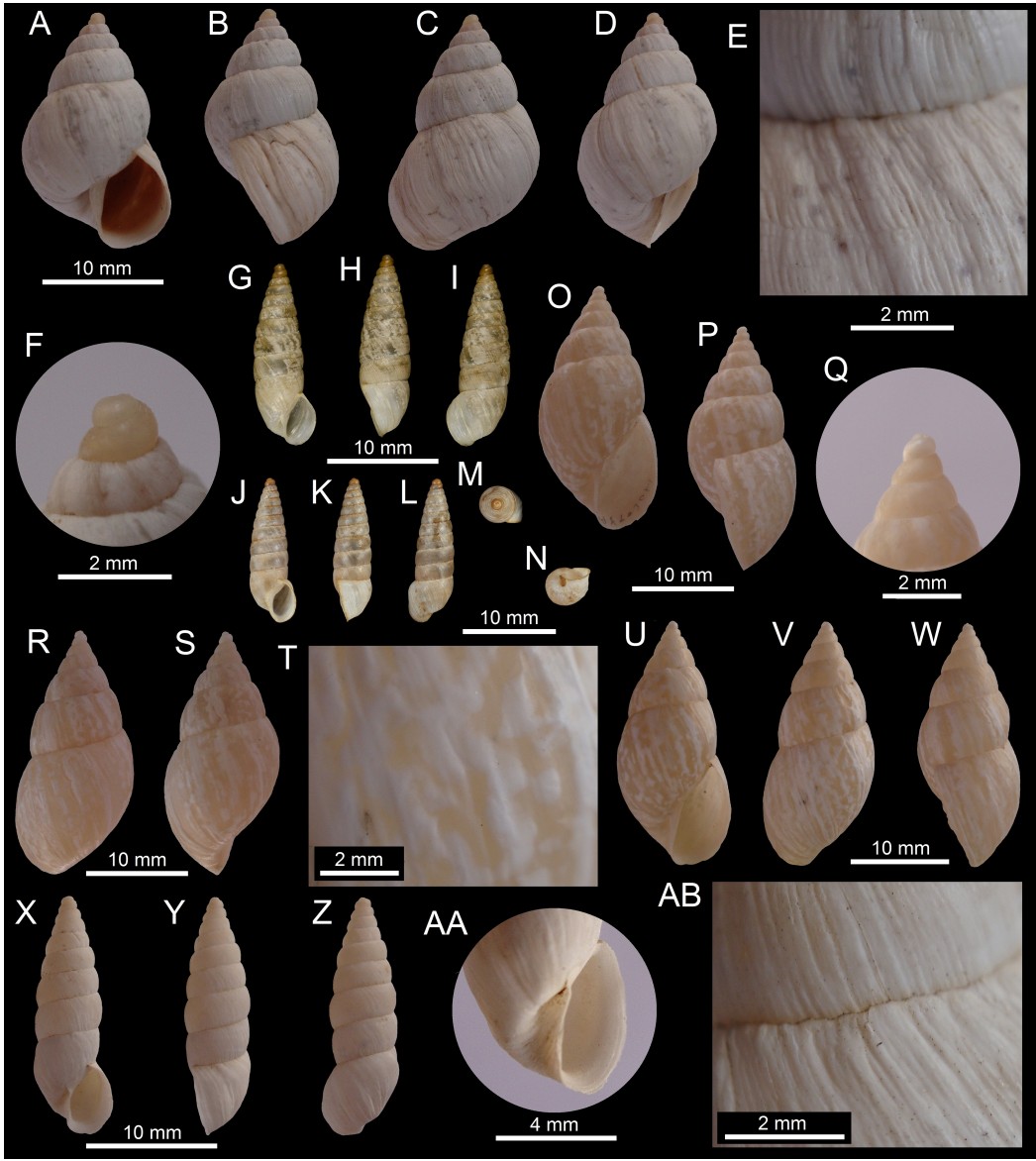

**Figure 6 Images of shells of *Bostryx* species.** *Bostryx huascensis* (Reeve, 1848) MPCCL 13102015M (*H*: 19.6 mm), (A) ventral view, (B) side view (lip), (C) dorsal view, (D) side view (umbilicus), (E) detail of sculpture and suture, (F) detail of protoconch; *Bostryx inaquosum Breure, 1978* HT USNM 537830 (*H*: 16.4 mm), (G) ventral view, (H) side view (lip), (I) dorsal view; *Bostryx inaquosum Breure, 1978* USNM 522565 (*H*: 15.7 mm), (J) ventral view, (K) side view (lip), (L) dorsal view, (M) apical view, (N) umbilical view; *Bostryx ireneae* sp. nov. HT MPCCL 13102015D (*H*: 24.1 mm), (O) ventral view, (P) side view (lip), (Q) detail of protoconch, (R) dorsal view, (S) side view (umbilicus), (T) detail of sculpture; *Bostryx ireneae* sp. nov. PT 1 MZUC 39616 (*H*: 25.0 mm), (U) ventral view, (V) dorsal view, (W) side view (lip); *Bostryx ischnus* (*Pilsbry, 1902*) MPCCL 13102015N (*H*: 18.3 mm), (X) ventral view, (Y) side view (lip), (Z) dorsal view, (AA) detail of last whorl and umbilicus, (AB) detail of sculpture and suture.

*Peronaeus philippii Rehder, 1945*: 104 (non *B. philippii* (*Pfeiffer, 1842*)). *Bostryx inaquosum Breure, 1978*: 92, pl. 10, Fig. 13. *Bostryx* (*Peronaeus*) *philippii Stuardo & Valdovinos, 1985*: 57; *Stuardo & Vega, 1985*: 135. *Bostryx* (*Peronaeus*) *philippi Valdovinos, 1999*: 150.

**Material examined:** Holotype; near Copiapó (27°21′29″S; 70°20′24″W), Región de Atacama, Chile (USNM 537830); additional specimen from the same location (USNM 522565). Examined from photographs only.

**Description** (**After** *Rehder, 1945*)**:** Shell moderately slender, thin, with 11 moderately convex whorls. Protoconch of two smooth whorls, with the last one-half whorl often showing microscopic crowded wavy riblets. Teleoconch generally smooth except for irregular growth wrinkles and occasional spiral impressed lines giving a malleated appearance to the surface. Shell corneous with irregular opaque-white maculations or streaks, though in the specimens with a heavier shell this mottling may be absent. Last whorl flattened between the periphery and the obtuse keel surrounding the moderately broad umbilicus. Aperture narrowly ovate, outer lip narrowly expanded, columellar lip dilated.

**Distribution:** This species was described by *Rehder (1945)* from hills near Copiapó, Región de Atacama. *Breure (1978)* renamed this species and recorded a new locality at the Quebrada de la Chimba, 10 km N of Antofagasta, Región de Antofagasta, in 350 m altitude.

**Remarks:** Intensive searches in the type locality around the city of Copiapó yielded no specimens of this taxon; further field collections in the Región de Antofagasta may be useful in order to assess the current presence of this species.

### *Bostryx ireneae* sp. nov.

Urn:lsid:zoobank.org:act:F54AAAB0-830B-41FA-B7D1-BC692083A590

(Figs. 6O–6W)

**Type material:** Holotype (MPCCL 13102015D; $H = 24.1$, $D = 11.3$, $LW = 16.8$, $HA = 11.2$, $WA = 5.9$, $NW = 7.0$), paratype 1 MZUC 39616 ($H = 24.2$, $D = 11.2$, $LW = 16.4$, $HA = 11.0$, $WA = 6.1$, $NW = 7.5$), paratype 2 MZUC 39609 ($H = 24.8$, $D = 10.8$, $LW = 16.9$, $HA = 11.4$, $WA = 6.0$, $NW = 7.5$), paratype 3 JFACL TG019 ($H = 26.9$, $D = 11.6$, $LW = 17.3$, $HA = 11.9$, $WA = 6.4$, $NW = 7.5$). All material from type locality.

**Type locality:** Chañaral de Aceituno (29°01′35″S; 71°26′20″W), Comuna de Freirina, Región de Atacama, Chile, coll. and leg. Ricardo Catalán G, February, 2006. All the specimens collected in sand dunes and sandy areas.

**Diagnosis:** A *Bostryx* species with an acute apex, characterized by a prominent last whorl of about 0.67 H, with a pattern of opaque white and tan lines on a slender, spindle-shaped, malleated translucent shell. Small umbilicus.

**Description:** Shell up to 26.9 mm, around 2.3 times as long as wide, fusiform, narrowly umbilicated, with convex whorls, rather stout. Translucent tan shell, with irregularly spaced axial streaks of various shades of opaque white. Shiny surface, with indistinct growth striae and malleation. Protoconch smooth, relatively large, somewhat bigger than first teleoconch whorl. Apex acute. Whorls 7.0–7.5, convex, a little angulated near

the sutures; last whorl compressed below. Aperture elongate-ovate, around 1.8 times as long as wide; *HA* around 0.45 H. Peristome thin, outer lip sharp and simple. Columellar margin straight and folded, a little receding, dilated above. Soft parts unknown.

**Distribution:** Only known from type locality.

**Etymology:** Named in honour of Irene Garrido, Coquimbo, Chile.

**Remarks:** This species differs from most other *Bostryx* species found in Chile by its elongated spindle-shaped and almost translucent shell. The only species which can be compared with *B. ireneae* is *Bostryx affinis* (Broderip, 1832), a species recorded only in Mejillones (23°00′S; 70°15′W), Región de Antofagasta (Pilsbry, 1896). The new species differs from *B. affinis* by having more convex spire whorls, a much larger last whorl, a smaller *HA/H* ratio, a straight and receding columella, and in having a sharp lip. *Bostryx breurei* sp. n. is somewhat similar to *B. ireneae* n. sp., however *B. ireneae* has a larger, stouter, shinier shell, with a larger spire, without the spiral sculpture characteristic of *B. breurei*.

***Bostryx ischnus*** (*Pilsbry, 1902*)

(Figs. 6X–6AB)

*Bulimus terebralis Pfeiffer, 1842*: 187 (non *B. terebralis* Bruguière, 1789); (Reeve, 1848), pl. 14, Fig. 79; Hupé *in* Gay, 1854: 115, pl. 3, Fig. 9. *Bulimulus (Peronaeus) terebralis* Pilsbry, 1896: 142, pl. 45, Fig. 30. *Bulimulus (Peronaeus) ischnus Pilsbry, 1902*: lxxi. *Peronaeus ischnus*: *Rehder, 1945*: 104. *Bostryx (Peronaeus) ischnus Stuardo & Valdovinos, 1985*: 57; *Stuardo & Vega, 1985*: 134; *Valdovinos, 1999*: 150. *Bostryx ischnus*: Breure & Ablett, 2014: 191, Figs. 2M–2N, L60.

**Material examined:** El Morro (27°08′43″S; 70°55′42″W), Comuna de Caldera, Región de Atacama, Chile, coll. by JF Araya, August 2012 to December 2012 (MZUC 39615, 2 sppm; MPCCL 13102015N, 1 spm). Measurements of illustrated spm (in mm): Figs. 6X–6AB MPCCL 13102015N ($H = 18.3$, $D = 6.1$, $LW = 8.2$, $HA = 5.4$, $WA = 3.4$, $NW = 10$).

**Description (After Pilsbry, 1896):** Shell rimate perforate, with the large umbilical area defined by an angle; subulate; whitish, becoming bluish-brown above. Surface lusterless, irregularly wrinkle-striate, the striae somewhat cut into granules by spiral impressed lines which are generally more prominent above. Spire tapering from the last whorl to the blunt, smooth, brownish-corneous apex. Whorls about 10, nearly flat, the last cylindrical, obliquely truncated below by a blunt but projecting angle, oner which the riblets do not pass, and which defines the umbilical tract. Aperture one-fourth the altitude, oblique, ovate, white inside. Outer lip acute, expanded and thickened within; columellar lip dilated above, vaulting over the minute perforation.

**Distribution:** *Stuardo & Vega (1985)* cited this species from Coquimbo, Región de Coquimbo and from Paposo, Región de Antofagasta. The species reported upon here fill the apparent distributional gap between those localities.

**Remarks:** This species is distinguishable from other similar elongated species due to the sharp basal keel around the umbilical area and the expanded outer lip (Fig. 6AA).

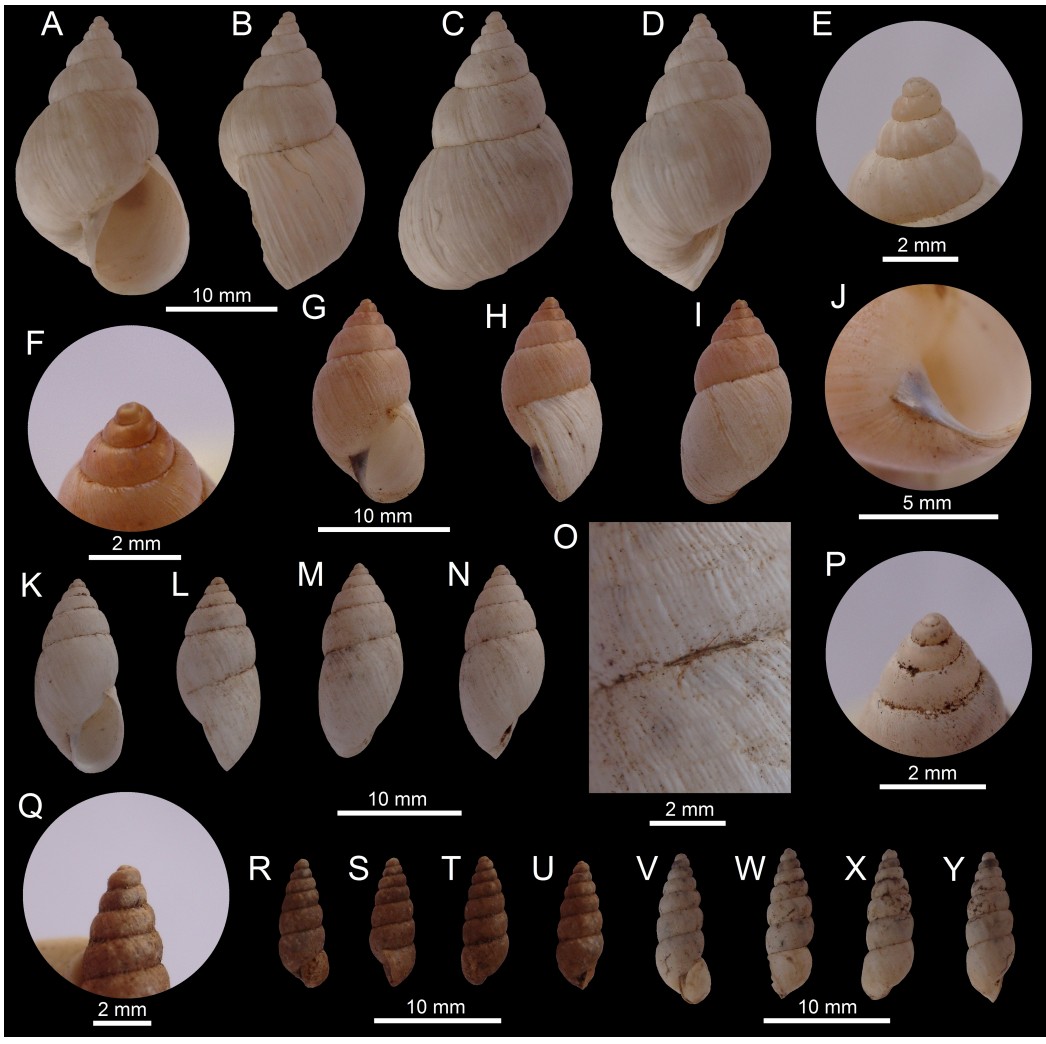

**Figure 7 Images of shells of *Bostryx* species.** *Bostryx mejillonensis* (Pfeiffer, 1857) MPCCL 13102015G (*H*: 25.2 mm), (A) ventral view, (B) side view (lip), (C) dorsal view, (D) side view (umbilicus), (E) detail of protoconch and early whorls; *Bostryx pruinosus* (Sowerby I, 1833) MZUC 39610 (*H*: 15.2 mm), (F) detail of protoconch and early whorls, (G) ventral view, (H) side view (lip), (I) dorsal view, (J) detail of columella; *Bostryx pruinosus* (Sowerby I, 1833) MPCCL 13102015F (*H*: 15.8 mm), (K) ventral view, (L) side view (lip), M: dorsal view, (N) side view (umbilicus), (O) detail of sculpture and suture, (P) detail of protoconch and early whorls; *Bostryx pumilio* (Rehder, 1945) MZUC 39614 (*H*: 10.2 mm), (Q) detail of protoconch an early whorls, (R) ventral view, (S) side view (lip), (T) dorsal view, (U) side view (umbilicus); *Bostryx pumilio* (*Rehder, 1945*) MPCCL 13102015B (*H*: 12.0 mm), (V) ventral view, (W) side view (lip), (X) dorsal view, (Y) side view (umbilicus).

Juvenile specimens of this species have a well-defined carinated last whorl, and are very similar to other elongate *Bostryx* species found in the area.

### *Bostryx mejillonensis* (Pfeiffer, 1857)

(Figs. 7A–7E)

*Bulimus mejillonensis* Pfeiffer in *Pfeiffer & Dunker, 1857*: 230; *Philippi, 1860*: 165, pl. 7, Figs. 10A–C; *Hidalgo, 1870*: 83. *Bulimulus* (*Lissoacme*) *mejillonensis* Pilsbry, 1896:

177, pl. 48, Figs. 11–14. *Bostryx (Lissoacme) mejillonensis* Stuardo & Valdovinos, 1985: 57; Stuardo & Vega, 1985: 135. *Bostryx (Peronaeus) mejillonensis* Valdovinos, 1999: 150. *Bostryx mejillonensis* Breure, 1978: 102; Breure & Ablett, 2014: 119, Fig. 8C, L37iii.

**Material examined:** Punta Frödden (26°56′S; 70°47′W) and Obispito (26°44′S; 70°42′W), Comuna de Caldera, Región de Atacama, Chile; coll. and leg. by Marta Araya, June 17, 2012 (MPCCL 13102015G, 1 specimen). Measurements of illustrated spm (in mm): Figs. 7A–7E MPCCL 13102015G ($H = 25.2$, $D = 15.7$, $LW = 18.7$, $HA = 12.2$, $WA = 9.8$, $NW = 6.0$)

**Description (After Pilsbry, 1896):** Shell oblong-ovate, solid; opaque-white, with irregular, interrupted corneous streaks, or uniform white. Surface lusterless, with irregular growth-wrinkles (and sometimes showing some faint incised spirals, or coarse malleation). Spire conic, with convex outlines, apex somewhat mamillar, whorls 6, convex. Aperture slightly less than half the length of shell, long ovate; outer lip sharp, with a broad and rather heavy internal thickening. Columella slightly concave, columellar lip dilated; parietal callus moderately heavy, white. Umbilicus narrow.

**Distribution:** *Stuardo & Vega (1985)* cite this species from Paposo and Mejillones, both localities at the Región de Antofagasta. This is the southernmost record for the taxon.

**Remarks:** This species has a small, sturdier whitish shell, with a thick lip (Pilsbry, 1896).

### *Bostryx pruinosus* (Sowerby I, 1833)

(Figs. 7F–7P)

*Bulinus pruinosus* Sowerby I, 1833: 36. *Bulimus pruinosus* Reeve, 1848: Pl. 20, f. 120. *Bulimulus (Lissoacme) pruinosus* Pilsbry, 1896: 175, pl. 1, figs 4–5. *Bostryx (Lissoacme) pruinosus* Stuardo & Valdovinos, 1985: 57; Stuardo & Vega, 1985: 135. *Bostryx (Peronaeus) pruinosus* Valdovinos, 1999: 151. *Bostryx pruinosus* Breure & Ablett, 2014: 158, Figs. 7F, L48vi.

**Material examined:** Caleta Pajonales (27°43′S; 71°02′W), Comuna de Copiapó, Región de Atacama, Chile; coll. by Ricardo Catalán G (RCGCL, unnumbered, 36 sppm); Aguas Verdes (26°52′S; 70°48′W), Comuna de Caldera, Región de Atacama, Chile, September 25, 2012; coll. by JF Araya (JFACL TG017, 1 specimen). Measurements of illustrated sppm (in mm): Figs. 7F–7J (MZUC 39610) ($H = 15.2$, $D = 8.3$, $LW = 11.6$, $HA = 7.0$, $WA = 6.0$, $NW = 5.0$); Figs.7K–7P (MPCCL 13102015F ($H = 15.8$, $D = 7.3$, $LW = 11.3$, $HA = 6.6$, $WA = 4.5$, $NW = 5.0$).

**Description (After Pilsbry, 1896):** Shell umbilicated, ovate-conic, rather solid but thin; corneous, flecked and streaked with white, or opaque white flecked and streaked with corneous, the latter predominating on spire, earlier whorls smooth, corneous. Surface smoothish, with wrinkles of growth, and above the periphery cut into spiral series of long granules by spiral lines. Spire conic, the apex obtuse, smooth. Whorls 5 and a half, slightly convex, the sutures more or less crenulated. Aperture half the total length (more or less), slightly oblique, ovate, white within; outer lip unexpanded, arcuate; columellar lip well dilated above.

**Distribution:** *Stuardo & Vega (1985)* cited this species from Cobija, Region of Antofagasta. This is the southernmost record for this species.

**Remarks:** This species has a distinctive sturdy but thin sub-cylindrical shell with a large last whorl. The large aperture and general proportions of the shell distinguish *B. pruinosus* from all the other *Bostryx* species found in Chile.

***Bostryx pumilio*** (*Rehder, 1945*)

(Figs. 7Q–7Y)

*Bulimus nanus* Reeve, 1849: 330, pl. 79, Fig. 585 (non *B. nanus* Lamarck, 1804). *Bulimulus* (*Peronaeus*) *nanus* Pilsbry, 1896: 141, pl. 45, Fig. 22. *Peronaeus pumilio Rehder, 1945*: 106. *Bostryx pumilio Breure, 1978*: 116. *Bostryx* (*Peronaeus*) *pumilio Stuardo & Valdovinos, 1985*: 57; *Stuardo & Vega, 1985*: 135; *Valdovinos, 1999*: 150. *Bostryx nanus*: *Köhler, 2007*:133, Fig. 28.

**Material examined:** El Morro (27°08′43″S; 70°55′42″W), Comuna de Caldera, Región de Atacama, Chile, August 2012 to January 2013; coll. by JF Araya (MZUC 39614, 5 specimens; MPCCL 13102015B, 3 specimens; RMNH.MOL 329667, 4 specimens). Measurements of illustrated sppm (in mm): Figs. 7O–7U MZUC 39614 ($H = 10.2$, $D = 4.0$, $LW = 5.0$, $HA = 3.0$, $WA = 2.2$, $NW = 8.0$); Fis.7V–7Y MPCCL 13102015B ($H = 12.0$, $D = 4.4$, $LW = 5.9$, $HA = 3.6$, $WA = 2.7$, $NW = 9.0$).

**Description (after** *Breure, 1978*): Shell small ($H$ up to 12 mm), rather elongated, thin and umbilicated. Eight convex, very finely striated whorls, striae slightly plicated beneath the sutures. Protoconch smooth but with fine spiral lines and low axial wrinkles under magnification. Color pale ash or corneous, obscurely marked here and there with light brown streaks, apex brown. Aperture small, subovate, angular at the basal margin; peristome thin and simple. Columelar margin dilated above.

**Distribution:** This is the first record after its description to give an actual locality for this species, which has been listed simply as "Chile" by all the precedent authors (*Reeve, 1848–1850*; *Pilsbry, 1895–1896*; *Stuardo & Vega, 1985*; *Köhler, 2007*).

**Remarks:** *Rehder (1945)* proposed the new name *pumilio* as *B. nanus* Reeve, 1849 was preoccupied by *B. nanus* Lamarck, 1804. This is a somewhat common species (one of the smallest *Bostryx* species from Chile, with very small shells, up to 12.0 mm) found in numbers at one of the locations under study (El Morro), always buried beneath roots and under large rocks.

***Bostryx pupiformis*** (**Broderip & Sowerby I, 1832**)

(Figs. 8A–8I)

*Bulinus pupiformis* Broderip & Sowerby I, 1832: 105; Reeve, 1848: pl. 14, Fig. 85; Hupé *in* Gay 1854: 114, pl. 2, Fig. 6; *Hidalgo, 1870*: 99. *Bulimulus* (*Peronaeus*) *pupiformis* Pilsbry 1896: 138, Pl. 45, Figs. 9–10. *Peronaeus pupiformis Rehder, 1945*: 104. *Bostryx* (*Peronaeus*) *pupiformis Stuardo & Valdovinos, 1985*: 57; *Stuardo & Vega, 1985*: 135; *Valdovinos, 1999*: 150. *Bostryx pupiformis Neubert & Janssen, 2004*: 226, Taf. 9, Fig. 93; *Köhler, 2007*: 133, Fig. 31.

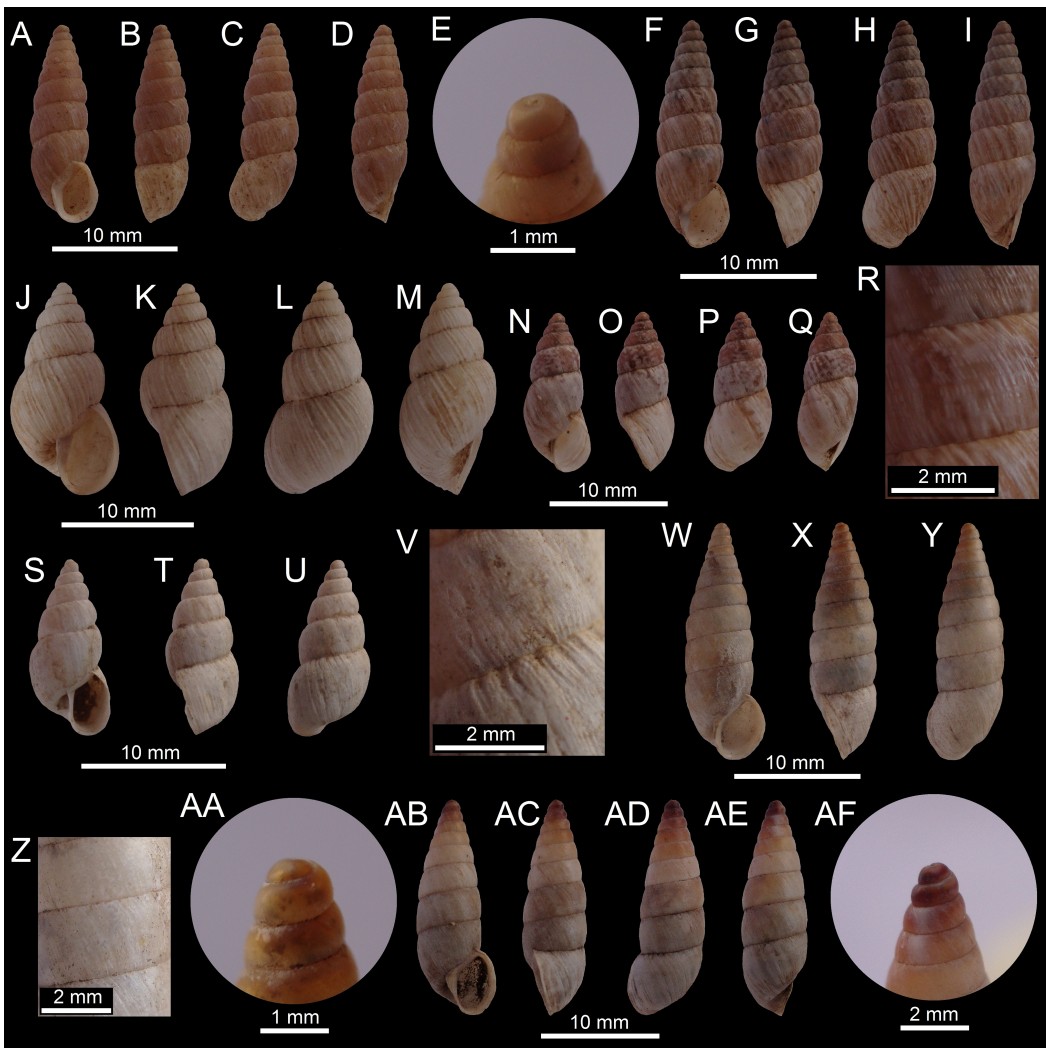

**Figure 8 Images of shells of *Bostryx* species.** *Bostryx pupiformis* (Pfeiffer, 1857) MZUC 39613 (*H*: 25.2 mm), (A) ventral view, (B) side view (lip), (C) dorsal view, (D) side view (umbilicus), (E) detail of protoconch and early whorls; *Bostryx pupiformis* (Sowerby I, 1833) MPCCL 13102015C (*H*: 15.2 mm), (F) ventral view, (G) side view (lip), (H) dorsal view, (I) side view (umbilicus); *Bostryx pustulosus* (Sowerby I, 1833) RCGCL unnumbered (*H*: 15.8 mm), (J) ventral view, (K) side view (lip), (L) dorsal view, (M) side view (umbilicus); *Bostryx rhodacme* (Rehder, 1945) MZUC 39612 (*H*: 10.2 mm), (N) ventral view, (O) side view (lip), (P) dorsal view, (Q) side view (umbilicus), (R) detail of sculpture and sutures; *Bostryx umbilicaris* (Rehder, 1945) RCGCL unnumbered (*H*: 12.0 mm), (S) ventral view, (T) side view (lip), (U) dorsal view, (V) detail of sculpture and sututre; *Bostryx valdovinosi* sp. nov. HT MPCCL 13102015H (*H*: 12.0 mm), (W) ventral view, (X) side view (lip), (Y) dorsal view, (Z) detail of sculpture and suture. *Bostryx valdovinosi* sp. nov. PT 1 MZUC 39611 (*H*: 12.0 mm), (AA) detail of protoconch and early whorls, (AB) ventral view, (AC) side view (lip), (AD) dorsal view, (AE) side view (umbilicus), (AF) detail of protoconch and early whorls.

**Material examined:** El Morro (27°09′34″S; 70°56′07″W), Comuna de Caldera, Región de Atacama, Chile, November 23, 2012; coll. by JF Araya (MZUC 39613, 2 sppm; MPCCL 13102015C, 1 sppm; RMNH.MOL 329668, 2 sppm). Measurements of illustrated sppm (in mm): Figs. 8A–8E MZUC 39613 (*H* = 15.9, *D* = 5.4, *LW* = 7.5, *HA* = 5.0, *WA* = 3.8,

$NW = 10$); Figs. 8F–8I MPCCL 13102015C ($H = 16.8$, $D = 5.5$, $LW = 7.7$, $HA = 4.9$, $WA = 3.7$, $NW = 9.5$).

**Description (after Pilsbry, 1896):** Shell rimate-perforate, long, tapering above, cylindrical below, rather solid; whitish, the earlier whorls blackish-orange or corneous, or entire shell corneous with narrow irregular white streaks. Surface smoothish, the growth-lines irregular, more prominent at sutures, and there is some superficial malleation throughout. Whorls 10-11, weakly convex, the last three of about the same diameter, those above tapering to a slightly mamillar, obtuse, glossy apex; last whorl tapering and somewhat compressed toward the base with a small umbilical excavation. Aperture slightly over one-fourth the total altitude, oblique, ovate; peristome obtuse, the outer lip regularly arcuate, distinctly expanded or spreading below; columellar lip expanded; ends connected by a white parietal callus.

**Distribution:** Pilsbry (1896) cited this species from Huasco, Región de Atacama and from Coquimbo, Región de Coquimbo. The specimens here constitute the northernmost record for this species.

**Remarks:** This is one of the most abundant elongated *Bostryx* species. It is a common species at the El Morro; its shells are usually found scattered among large rocks and fresh specimens were collected buried in sand under and among the roots of dried herbs.

### *Bostryx pustulosus* (Broderip, 1832)

(Figs. 8J–8M)

*Bulinus pustulosus* Broderip & Sowerby I, 1832: 105. *Bulimus pustulosus* Hupé in Gay, 1854: 112, Malacologia, lám. 2. Figs. 4, 4a–f; Reeve, 1848: Pl. 20, Fig. 127; *Hidalgo, 1870*: 90. *Bulimulus (Peronaeus) pustulosus* Pilsbry, 1896: 153, pl. 46, Fig. 58. *Peronaeus pustulosus Rehder, 1945*: 106. *Bostryx (Peronaeus) pustulosus Stuardo & Valdovinos, 1985*: 57; *Stuardo & Vega, 1985*: 135; *Valdovinos, 1999*: 150; *Köhler, 2007*: 133, Fig. 32. *Bostryx pustulosus Neubert & Janssen, 2004*: 226, Taf. 9, Fig. 94.

**Material examined:** Hills near Pan de Azucar (26°10′01″S; 70°39′22″W), Comuna de Chañaral, Región de Atacama, Chile; coll. and leg. by Ricardo Catalán G (RCGCL unnumbered, 5 sppm). Measurements of illustrated spm (in mm): Fig. 8J–8M RCGCL unnumbered ($H = 16.2$, $D = 8.3$, $LW = 10.9$, $HA = 6.8$, $WA = 4.8$, $NW = 6.0$)

**Description (after Pilsbry, 1896):** Shell umbilicate-rimate, oblong-conical, rather solid, calcareous. White or pale brownish, often with some scattered corneous-brown dots, the earlier two whorls corneous. Surface lusterless, rudely closely and irregularly plicate-striate, with stronger folds at rather wide but unequal intervals; the longitudinal folds and striae cut into rather coarse granules by incised spirals, unequally developed; the sculpture weaker above, absent on the smooth apical whorls. Spire conic; whorls 6 to 6 and a half, convex, the last with an ample umbilical excavation, but only a minute perforation. Aperture subovate, contained 2.16–2.3 times in altitude of shell, whitish or brown inside; peristome slightly thickened within, the outer lip regularly arcuate, not expanded, columellar lip dilated above, ends of the lip approaching.

**Distribution:** *Stuardo & Vega (1985)* cited this species as only occurring in Huasco. This is the northernmost record for the species.

**Remarks:** A conspicuous, slightly elongated species with a convex conic spire and a somewhat pustulose sculpture, consisting of longitudinally granulated striae crossed by spiral grooves. The latter characteristic sets this species apart from all other known *Bostryx* species in Chile.

### *Bostryx rhodacme* (*Pfeiffer, 1843*)

(Figs. 8N–8R)

*Bulimus rhodacme Pfeiffer, 1843*: 50; Reeve, 1848: pl. 14, Fig. 77; Hupé in Gay, 1854: 113. *Bulimulus rhodacme* Pilsbry, 1896: 152, pl. 46, Fig. 54. *Peronaeus rhodacme Rehder, 1945*: 106. *Bostryx (Peronaeus) rhodacme Stuardo & Valdovinos, 1985*: 57; *Stuardo & Vega, 1985*: 135; *Valdovinos, 1999*: 150; *Breure, 2013*: 42, Fig. 14B, 14 ii.

**Material examined:** Comuna de Vallenar (28°34′S; 70°45′W), Región de Atacama, Chile, December 2010, leg. by Ricardo Catalán G (RCGCL unnumbered, 23 sppm; MZUC 39612, 2 sppm). Measurements of illustrated spm (in mm): Figs. 8N–8R MZUC 39612 ($H = 13.6, D = 5.8, LW = 8.8, HA = 4.8, WA = 3.5, NW = 6.0$).

**Description (after Pilsbry, 1896):** Shell openly rimate, narrowly long-ovate, rather solid. White, with irregular pellucid or pink maculation, often suffused with a roseate blush on the spire, sometimes unicolored white. Surface lusterless, with rude, irregular growth-striae, sparsely decussated, or bearing granules in spiral series; spiral sculpture of very fine lines. Spire conic, with slightly convex outlines, the apex obtuse. Whorls 6, the upper ones very convex with deep sutures, the lower two less convex, suture slightly and irregularly crenulate; the apical first and a half whorl smooth, corneous or roseate. Last whorl not compressed below. Aperture over one-third the height of shell, long-oval, white within; peristome not expanded, slightly thickened within, the columellar margin dilated above.

**Distribution:** Pilsbry (1896) cited this species only from Freirina, near Huasco, in the same area where this species was recorded in the present work.

**Remarks:** This species is recognized by its small and whitish-pinkish variegated shell with a long last whorl, high spire and a lustrous shell, faintly sculptured by strong growth striae.

### *Bostryx umbilicaris* (*Souleyet, 1842*)

(Figs. 8S–8V)

*Bulimus umbilicaris Souleyet, 1842*: 102; *Souleyet, 1852*: 513, pl. 29, Figs. 12–15. *Bulimulus (Ataxus) umbilicaris* Pilsbry, 1896: 130, pl. 44, Figs. 87–88. *Bostryx (Ataxus) umbilicaris Breure, 1975*: 1140, pl. X, Fig. 5; *Stuardo & Valdovinos, 1985*: 57; *Stuardo & Vega, 1985*: 134; *Valdovinos, 1999*: 150. *Bostryx umbilicaris Breure, 1978*: 138, Figs. 225–229.

**Material examined:** Caleta Pajonales (27°43′S; 71°02′W), Comuna de Copiapó, Región of Atacama, Chile, December 2010, leg. Ricardo Catalán G. (RCGCL unnumbered, one

spm). Measurements of illustrated spm (in mm): Figs. 8S–8V RCGCL unnumbered ($H = 12.3$, $D = 5.7$, $LW = 7.7$, $HA = 4.4$, $WA = 2.9$, $NW = 6.0$).

**Description (after Pilsbry, 1896):** Shell with ample, well-like umbilicus, long-conic; thin but rather solid; opaque, white or flesh-tinted; smooth except for slight growth-lines usually indistinctly crimped in spiral order, shining. Spire attenuated above, the earlier whorls smooth, corneous, mamillar; sutures slightly plicated. Whorls 7, slightly convex but somewhat flattened at the periphery; regularly widening, the last not deflexed, narrowed toward the base, forming a narrowly rounded ridge around the very large umbilicus, which penetrates well-like to the apex. Aperture small, oblique, narrowly oblong; peristome white, subcontinuous, in contact with the body-whorl only for an extremely short distance above; outer and basal lips narrowly expanded, columellar lip straighter, dilated.

**Distribution:** *Stuardo & Vega (1985)* cited this species from Cobija, Región de Antofagasta. This is the southernmost record for this species.

**Remarks:** This small species is clearly identified by the small and elongated whitish shell with slightly flattened whorls, a somewhat angulated last whorl and a large umbilicus. This seems to be a rare species, with only a single empty shell specimen found.

*Bostryx valdovinosi* **sp. nov.**

Urn:lsid:zoobank.org:act:3266D2D9-888B-4BB3-84F7-0E09ACB73F9A

(Figs. 8W–8AF)

**Type material:** Holotype MPCCL XXXX ($H = 18.2$, $D = 6.4$, $LW = 8.2$, $HA = 6.0$, $WA = 4.2$, $NW = 9.5$). paratype 1 MZUC 39611 ($H = 18.8$, $D = 6.2$, $LW = 8.3$, $HA = 5.8$, $WA = 4.3$, $NW = 10$), paratype 2 JFACL TG016 ($H = 15.6$, $D = 5.4$, $LW = 7.2$, $HA = 5.0$, $WA = 3.7$, $NW = 9.0$). Coll. by JF Araya, 17 december 2010; all material from type locality.

**Type locality:** SE side of El Morro (27°08′43″S; 70°55′42″W, 194), Comuna de Caldera, Región de Atacama, Chile. All the specimens were collected among the roots of dry herbs and under rocks.

**Diagnosis:** An elongated-turrited *Bostryx* species characterized by a comparatively wide shell with a dark coloured protoconch, rimate umbilicus, an ovate straight aperture and a narrowly expanded peristome.

**Description:** Shell elongated-turrited, stout, corneous or opaque white, height up to 18.8 mm, $H/D$ about 0.34, perforated, with convex whorls. Surface shiny with tenuous growth lines. Upper whorls more convex and darker in colour. Protoconch reddish-brown, smooth and comparatively small. Whorls 9.0–10.0, convex, suture impressed. Aperture ovate, 0.27 times the total height. Peristome narrowly expanded. Columellar margin concave, sharp, and dilated above, reflected over the umbilical perforation. Outer lip sharp, white. Interior of aperture pinkish or brownish. Soft parts unknown.

**Distribution:** Only known from type locality.

**Etymology:** This species is named after Claudio Valdovinos (Universidad de Concepción), in recognition of his works dealing with the Chilean Mollusca.

**Remarks:** This rare species is slightly similar to *B. inaquosum*, from which it differs in having a larger, more opaque shell with a less slanted and wider aperture. It can also be compared to *B. pupiformis*, which have a smaller and more elongated aperture, a much more tenuous parietal callus and a comparatively smaller last whorl. *Bostryx anachoreta* (*Pfeiffer, 1856*) differs in having a larger and more elongated shell with a convex-turrited spire (almost concave in the new species), and a larger and narrower aperture.

## DISCUSSION

Most of the original descriptions of *Bostryx* species occurring in Chile have been based solely on conchological characters, and only a few of them have been the subject of additional studies including soft parts (*Breure, 1978*; *Valdovinos & Stuardo, 1988*). Thus, *Bostryx* species have been traditionally located in several subgenera in reference to the morphology of the shell; *Bostryx* s.s. *Troschel, 1847* represented by minute, almost uncoiled shells with a large umbilicus and somewhat flattened whorls (with a single species in Chile: *Bostryx holostoma* (*Pfeiffer, 1856*)); *Ataxus Albers, 1850*, characterized by umbilicated shells (represented solely by *B. umbilicaris*); *Peronaeus Albers, 1860* represented by elongated or turrited shells, with small apertures (e.g., *B. inaquosum*, *B. ischnus*, *B. pumilio*, *B. pupiformis*,); *Lissoacme* Pilsbry, 1896 represented by obese-ovate shells, often with large protoconchs (e.g., *B. albicans*, *B. erythrostomus*, *B. huascensis*, *B. mejillonensis*), and *Platybostryx* Pilsbry, 1896, with depressed and often carinated shells (represented in Chile by *B. eremothauma*). This characterization was changed by *Rehder (1945)* who elevated the subgenus *Peronaeus* to genus level, however this was not followed by subsequent works dealing with Chilean terrestrial land snails (*Breure, 1978*; *Stuardo & Valdovinos, 1985*; *Stuardo & Vega, 1985*; *Valdovinos, 1999*). In this regard I am here following *Breure (1978)* and treating all *Bostryx* species as belonging to *Bostryx* s. l.

The Region of Atacama has recently been the subject of several works dealing with its invertebrate fauna, which has uncovered new records and new species endemic to the area (*Osorio, 2012*; *Araya, 2013*; *Araya & Aliaga, 2015b*; *Araya & Araya, 2015a*; *Araya & Araya, 2015b*; *Reiswig & Araya, 2014*; *Collado, 2015*; *Labrín, Guzmán & Sielfeld, 2015*). Most of the taxa discussed in this study were also recorded in often narrow distributions (Table 2), or single localities along the coastal desert. This endemism is alike to that of other bulimid species living in higher altitudes in Peru (*Breure & Mogollón, 2010*; *Breure & Neubert, 2008*). In particular, it has been documented that *Bostryx* snails undergo long-term estivation and that their communities are subjected to the *El Niño-Southern Oscillations* (ENSO) in the *Lomas* habitat of coastal Peru (*Ramírez et al., 1999*). It is very possible that the same circumstances affect the *Bostryx* communities along the hyper arid coastal areas of the Región de Antofagasta and of the Región de Atacama, where the abundance of shells was correlated with the El Niño phenomena with much more fresh specimens, including the only living snails, found in humid years (JF Araya, unpublished obs., 2015). It was noticed also that there is some relationship between the abundance of

**Table 2 Geographic distribution for the species under study.** Distribution range of bulimulid species found in the Región de Atacama, based on *Breure (1978)*, *Stuardo & Vega (1985)*, *Stuardo & Vargas-Almonacid (2000)*, *Köhler (2007)* and this study.

| Species | Distribution | References |
|---|---|---|
| *Bostryx albicans* (Broderip, 1832) | Caldera (27°04′S; 70°49′W) to Coquimbo (29°57′S; 71°20′W). | *Stuardo & Vega, 1985* and this study |
| *Bostryx ancavilorum* sp. nov. | Aguas Verdes (26°52′S; 70°48′W), Caldera. | This study |
| *Bostryx breurei* sp. nov. | Quebrada del León (26°57′S; 70°44′W), Caldera. | This study |
| *Bostryx calderaensis* sp. nov. | Caldera (27°04′S; 70°49′W). | This study |
| *Bostryx erythrostomus* (*Sowerby, 1833*) | Caldera (27°04′S; 70°49′W) to Coquimbo (29°57′S; 71°20′W). | *Stuardo & Vega, 1985* |
| *Bostryx huascensis* (Reeve, 1848) | La Virgen beach (27°21′S; 70°57′W) to Huasco (28°20′S; 71°15′W). | *Stuardo & Vega, 1985* and this study |
| *Bostryx inaquosum* *Breure, 1978* | La Chimba (23°32′17″S; 70°21′6″W), Antofagasta and Copiapó (29°01′35″S; 71°26′20″W). | *Breure, 1978* |
| *Bostryx ireneae* sp. nov. | Chañaral de Aceituno (29°01′35″S; 71°26′20″W), Freirina. | This study |
| *Bostryx ischnus* (*Pilsbry, 1902*) | Paposo (25°05′S; 70°25′W) to Coquimbo (29°57′S; 71°20′W). | *Stuardo & Vega, 1985* and this study |
| *Bostryx mejillonensis* (Pfeiffer, 1857) | Mejillones (23°00′S; 70°15′W) to Punta Fröden (26°56′S; 70°47′W) Caldera. | *Stuardo & Vega, 1985* and this study |
| *Bostryx pumilio* (*Rehder, 1945*) | El Morro hill (27°08′43″S: 70°55′42″W), Caldera. | This study |
| *Bostryx pupiformis* (Broderip, 1832) | El Morro hill (27°08′43″S; 70°55′42″W) Caldera to Coquimbo (29°57′S; 71°20′W). | *Stuardo & Vega, 1985* and this study |
| *Bostryx pruinosus* (*Sowerby, 1833*) | Cobija (22°33′S; 70°16′W) and Caleta Pajonales (27°43′S; 71°02′W) Freirina. | *Stuardo & Vega, 1985* and this study |
| *Bostryx pustulosus* (Broderip, 1832) | Pan de Azucar National Park (26°08′S; 70°39′W) to Coquimbo (29°57′S; 71°20′W). | *Stuardo & Vega, 1985* and this study |
| *Bostryx rhodacme* (*Pfeiffer, 1843*) | Freirina (28°30′S; 71°04′W) to Vallenar (28°34′S; 70°45′W). | Pilsbry (1896) and this study |
| *Bostryx umbilicaris* (*Souleyet, 1842*) | Cobija (22°33′S; 70°16′W) and Caleta Pajonales (27°43′S; 71°02′W), Freirina. | *Stuardo & Vega, 1985* and this study |
| *Bostryx valdovinosi* sp. nov. | El Morro (27°08′43″S; 70°55′42″W), Caldera. | This study |

shells and the presence of lichen communities in some specific areas; this is consistent with the observation that *Bostryx* species have been found to feed in lichen and algae (*Ramírez et al., 1999*). It seems that these snails avoid heat stress and desiccation by retreating beneath the ground surface during the day and being active only when the humidity is high and at night. This behaviour was observed in specimens maintained in captivity, according to *Gigoux (1932)*, and was confirmed during this work. This mode of living may answer why many shells are found in areas with no apparent vegetation, where some species may have living communities despite the aridity of the region. The vast predominance of white shells among the Chilean *Bostryx* species (nine species of a total of 17 species studied herein) may also represent a strategy for reducing solar radiation absorption and thereby reduce heat, as recorded for other desert snails living in similar conditions (*Schmidt-Nielsen, Taylor & Shkolnik, 1971*).

In summary, the family Bulimulidae includes 17 known species identified in the Región de Atacama (Table 2), making it one of the richest terrestrial snail regions in Chile. This study confirms the existence of several previously undiscovered species in a poorly studied part of the country and similar findings can be expected in other poorly surveyed areas, specially towards northern Chile. In this regard this work complements recent work dealing with non-bulimulid species found in the Región de Atacama (*Miquel & Araya, 2013*; *Araya, 2013*; *Araya & Catalán, 2014*). Considering the restricted distributions of most of the species here reviewed, I propose that several be considered as candidates for threatened species status, according to the IUCN Red list Categories and Criteria.

**Key for the bulimulid snails from the Atacama Region, based on shell characters of adult shells.**

| | | |
|---|---|---|
| **1.** | Shell very elongated or turrited....... | **2** |
| **1a.** | Shell obese-ovate to ovate-elongated or fusiform....... | **6** |
| **2.** | Shell medium sized ($H > 12$ mm, up to 28 mm), sculptured by fine growth lines....... | **3** |
| **2a.** | Shell elongate to turrited, very small ($H$ up to 11 mm), white or corneous, with convex whorls sculptured only by growth lines, sutures deep, aperture small, oval....... | ***Bostryx pumilio*** (*Rehder, 1945*) (Figs. 7V–7Y) |
| **3.** | Shell corneous, elongate-turrited, slightly large (up to 27 mm), tapering from the middle of the shell to the apex, outer lip simple....... | **4** |
| **3a.** | Shell plain white, elongate-turrited, slightly large (up to 28 mm), and tapering in all length, outer lip acute anteriorly, slightly expanded or flared below, umbilical area defined by a sharp angle in last whorl....... | ***Bostryx ischnus*** (*Pilsbry, 1902*) (Figs. 6X–6AB) |
| **4.** | Protoconch and earlier whorls corneous or pale white, shell decorated with irregular white maculations or streaks, last whorl flattened, with an obtuse keel around the umbilicus, aperture narrowly ovate, slightly slanted, outer lip slightly expanded....... | ***Bostryx inaquosum*** *Breure, 1978* (Figs. 6G–6N) |
| **4a.** | Protoconch and earlier whorls pale brown to reddish-brown ....... | **5** |
| **5.** | Shell tapering posteriorly, cylindrical below, sculptured by fine growth marks, aperture ovate-elongate (about 0.25 H)....... | ***Bostryx pupiformis*** (**Broderip, 1832**) (Figs. 8A–8I) |
| **5a.** | Shell with earlier whorls convex and darker, with a reddish-brown protoconch; last whorl large, with a slightly raised parietal callus, aperture ovate (about 0.27 H)....... | ***Bostryx valdovinosi*** **sp. nov.** (**Figs. 8W–8AF**) |
| **6.** | Shell sculptured only by growth lines, spiral lines absent or very faint....... | **7** |
| **6a.** | Shell sculptured by strong growth lines and spiral lines which create minute axial pustules, whorls slightly plicated near the deep sutures, aperture ovate, lip simple, size small ($H$ up to 17 mm)....... | ***Bostryx pustulosus*** (**Broderip, 1832**) (Figs. 8J–8M) |
| **7.** | Shell small ($H$ up to 14 mm), oblong-ovate, with a large last whorl ($LW > 0.5$ H)....... | **8** |
| **7a.** | Shell large ($H$ up to 32 mm), obese to fusiform, $HA > 0.3$ H, umbilicus deep....... | **9** |
| **8.** | Shell white or dull grayish, whorls slightly flattened, with an impressed and slightly crenulated suture, umbilicus very ample and deep, straight columellar lip, aperture oval, slightly narrow....... | ***Bostryx umbilicaris*** (*Souleyet, 1842*) (Figs. 8S–8V) |
| **8a.** | Shell corneous, thin, with a pink or reddish apex, shell decorated with sparse white and pink streaks or maculations, last whorl about 0.4 H....... | ***Bostryx rhodacme*** (*Pfeiffer, 1843*) (Figs. 8N–8R) |
| **9.** | Shell obese-ovate, solid, aperture white, pink or redish-brown, $HA > 0.5$ H, lip simple....... | **10** |
| **9a.** | Shell oblong-ovate to fusiform, thin....... | **13** |
| **10.** | Protoconch corneous, white or pinkish....... | **11** |

## ACKNOWLEDGEMENTS

I thank Ricardo Catalán (Gobierno Regional, Copiapó, Chile) for his help and access to examine and photograph specimens from his private collection; Abraham Breure (Naturalis Biodiversity Center, Leiden, the Netherlands) for helping with literature and for his suggestions on an early version of the manuscript; Eike Neubert (Naturhistorisches Museum der Burgergemeinde Bern, Bern, Switzerland) and Frank Köhler (Australian Museum, Sydney, Australia) for helping with literature; Jonathan Ablett (National History Museum, London, UK) and Robert Hershler (Smithsonian Institute, National Museum of Natural History, Washington DC, USA) for helping with images of type specimens from the NHMUK and the SI NMNH respectively; Adam Baldinger (Museum of Comparative Zoology, Harvard University, Cambridge, USA), for his help with images of *Bostryx* species deposited in the MCZ; Sergio Miquel (Museo Argentino de Ciencias Naturales Bernardino Rivadavia, Buenos Aires, Argentina) for suggestions on an early version of the manuscript; Daniel Geiger (Santa Barbara Museum of Natural History, Santa Barbara, USA) and Megan Paustian (Carnegie Museum of Natural History, Pittsburgh, USA) for useful suggestions and correction of the manuscript and Marta Araya (Caldera,

Chile) for her help in the field collections. I thank also the reviewers; Francisco J. Borrero (Carnegie Museum of Natural History, Pittsburgh, USA); Cleo Oliveira (Museu Nacional da Universidade Federal do Río de Janeiro, Río de Janeiro, Brasil), and Alejandra Rumi (Museo de La Plata, Universidad Nacional de La Plata, Buenos Aires, Argentina) for their helpful corrections, suggestions and comments, which greatly improved this manuscript.

### Funding
The author received no funding for this work.

### Competing Interests
The author declares there are no competing interests.

### Author Contributions
- Juan Francisco Araya conceived and designed the experiments, performed the experiments, analyzed the data, contributed reagents/materials/analysis tools, wrote the paper, prepared figures and/or tables, reviewed drafts of the paper.

### Data Availability
The research in this article did not generate any raw data.

### New Species Registration
The following information was supplied regarding the registration of a newly described species:
Zoobank
urn:lsid:zoobank.org:pub:3F2E9582-A5D0-431D-BC3E-3917DC812323
*Bostryx ancavilorum* sp. nov. urn:lsid:zoobank.org:act:79735125-EE09-4FEF-B1D1-E86810CF7372
*Bostryx breurei* sp. nov. urn:lsid:zoobank.org:act:312AEEF0-C859-4C26-BEBD-8C03CA664A10
*Bostryx calderaensis* sp. nov /urn:lsid:zoobank.org:act:F54AAAB0-830B-41FA-B7D1-BC692083A590
*Bostryx ireneae* sp. nov. urn:lsid:zoobank.org:act:68913DD6-DA4B-4A9F-AE82-92F7970B9B01
*Bostryx valdovinosi* sp. nov. urn:lsid:zoobank.org:act:3266D2D9-888B-4BB3-84F7-0E09ACB73F9A

### Supplemental Information
Supplemental information for this article can be found online at http://dx.doi.org/10.7717/peerj.1383#supplemental-information.

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

## FURTHER READING

**Sowerby GBI. 1833–1838.** *Conchological illustrations, or coloured figures of all the hitherto unfigured recent shells, Bulinus* [5]–8, 103 figs.