# Peer review of "The Bulimulidae (Mollusca: Pulmonata) from the Región de Atacama, northern Chile"

_PeerJ, doi:10.7717/peerj.1383_

## Round 0.1 · original submission · Minor Revisions

Dear Dr. Araya,

As you'll notice, each reviewer provided a series of small, but valuable, suggestions to the manuscript. Please include as much as possible of those suggestions, and provide a justification for those you disagree with or find unfeasible.

·

Basic reporting

No comments

Experimental design

No Comments

Validity of the findings

MINOR CHANGES

- In the abstract, after the “of the seventeen species found”, author should include that among the seventeen species recorded to Atacama desert in this manuscript, part were not found during the field surveys and in this case the records are based on literature/collection material.

- Should avoid using Keywords that are already in the title, as “Chile” or “new species”.

- Along all the manuscript, information about repository number for MZUC is missing. Must include it.

- Should follow a patter in the terminology, using the word “specimen” for shell and soft parts included; and “shell” when only empty shell (even the eroded ones) was found.

- “RCG” is an abbreviation used several times (lines 236, 415, 441, 464, 484) but there is no explanation about what “RCG” means in Material and Methods.




IMPORTANT CONSIDERATIONS

- Bostryx catalani sp. nov. is cited several times in the manuscript (in the remarks of Bostryx breurei sp. nov., Line 182; figured in Figure 3a-c; cited twice in Table 1), but it is not described in any moment, neither the rules of ICZN were followed in this case.

- Considering the number of species and characters raised, it is important to include an informative table comparing the species at the end of the manuscript.

- For many, if not most, of species studied here, there are no indication of the number of examined specimens or shells. Must include it. It is particularly important for the new species described. Describing new species based on just few specimens can be undesired since small variations as shell outline or colour may represent individual phenotypical variation instead of interspecific variation of characters.

- For some species is given an estimative of the number of analysed specimens (e.g. Bostryx erythrostoma, 30+ specimens). But “30+” means any number above 30 specimens, including 40, 50, 100...1000. Should give the exactly number, no excuse.

- Line 283: Author considered that Bostryx inaquosum may be an extinct species in the Region of Atacama since it was not found during the fields survey. Such affirmative can not be done, it is very premature, even for such extensive field survey. The only conclusion that can be done here is that no specimens of this taxa were found.

- In the “Distribution” topic for the previously known species examined in this manuscript, author usually considered only one or two references dealing with these species. Are those references the only studies concerning the species studied here or are those references the only studies concerning the Atacama desert (and also the species studied here)? There is a huge difference between both statements since if the second one is the case, the distribution of these species can be bigger than the restricted area cited here.

- Author gave descriptive morphometrical data for the examined species, but no statistical tests were performed to analyze the significance of such morphometric variation. Of course, it is out of the scope of this manuscript (which is already a big work and a great contribution), but author are encouraged to think about a second manuscript analysing such variation among the species from Atacama desert and how this variation can be related to the areas habited by each species.

Additional comments

It was a pleasure to review the manuscript, which represents a great contribution on a poorly studied malacological fauna. As all good taxonomic manuscript, it still need further improvements.

·

Basic reporting

The MS is valuable and constructive comments for the author are in the attached file,
under Attachments.

Experimental design

There is minimal experimental design that is relevant to this particular study, and the author had adhered to standard methodology, which is well described in the MS. Constructive comments for the author are in the attached file, under
Attachments.

Validity of the findings

The finding are valid, well described, and valuable.
Constructive comments for the author are in the attached file, under Attachments.

Additional comments

See constructive comments for the author in the attached file, under General comments to the author.

·

Basic reporting

While work is well presented, gathering valuable background information of the little-known group, the biggest problem is methodological work. To wit:
Data on the basis of which have not been reached to distinguish new species described (5 new), even has the number of individuals observed for each of these or to the other already described (in total one entities 17), not the measurements obtained are presented and either a basic and essential statistics.
I understand that working with material collection only counts with shells, but also I know that there are very few measures employed as many measurements and soft packages that analyze the specimens under traditional methodologies currently exist, reaching other shape analysis: geometric morphometry.
Furthermore, the author mentions that also had live material. Why he did not analyze soft tissue or radula?

Experimental design

Absent

Validity of the findings

It is essential at least one table of data to verify the descriptions.

Additional comments

The figures are insufficient. It would be worthwhile to add more details protoconch, eg SEM of new species and highlight the differences with the already described.
I also detected some minor bug in MS, but do not consider relevant in this instance dondese needs further modification of the paper

---

## Round 0.2 · accepted · Accept

Dear Dr. Araya,

I'm glad to see that you revised your manuscript according to the reviewer's comments and that all of the raised issues have been properly addressed. With respect to the abstract in Spanish, I'm not sure if this is allowed according to the journal's policies, but I'm sure they'll be in touch.